# Admixture of evolutionary rates across a butterfly hybrid zone

**Tianzhu Xiong[1]\*, Xueyan Li[2], Masaya Yago[3], James Mallet[1]**

[1]Department of Organismic and Evolutionary Biology, Harvard University, Cambridge, United States; [2]Kunming Institute of Zoology, Chinese Academy of Sciences, Kunming, China; [3]The University Museum, The University of Tokyo, Tokyo, Japan

**Abstract** Hybridization is a major evolutionary force that can erode genetic differentiation between species, whereas reproductive isolation maintains such differentiation. In studying a hybrid zone between the swallowtail butterflies *Papilio syfanius* and *Papilio maackii* (Lepidoptera: Papilionidae), we made the unexpected discovery that genomic substitution rates are unequal between the parental species. This phenomenon creates a novel process in hybridization, where genomic regions most affected by gene flow evolve at similar rates between species, while genomic regions with strong reproductive isolation evolve at species-specific rates. Thus, hybridization mixes evolutionary rates in a way similar to its effect on genetic ancestry. Using coalescent theory, we show that the rate-mixing process provides distinct information about levels of gene flow across different parts of genomes, and the degree of rate-mixing can be predicted quantitatively from relative sequence divergence ($F_{ST}$) between the hybridizing species at equilibrium. Overall, we demonstrate that reproductive isolation maintains not only genomic differentiation, but also the rate at which differentiation accumulates. Thus, asymmetric rates of evolution provide an additional signature of loci involved in reproductive isolation.

## Editor's evaluation

The authors leverage theory, simulations, and empirical population genomics to evaluate what are the consequences of differences in substitution rates in hybridizing species. This is a largely overlooked pheonomenon. This study highlights the issue and demonstrates that two hybridizing species of Papilio have differences in thir substitution rates. The work will be of interest to a large group of evolutionary biologists, especially those studying evolution at the whole-genome level.

\*For correspondence:
txiong@g.harvard.edu

**Competing interest:** The authors declare that no competing interests exist.

## Introduction

DNA substitution, the process whereby single-nucleotide mutations accumulate over time, is a critical process in molecular evolution. Both molecular phylogenetics and coalescent theory rely on observed mutations (*Yang and Rannala, 2012*; *Wakeley, 2016*), and so the rate of substitution/mutation is the predominant link from molecular data to information about the timing of past events (*Bromham and Penny, 2003*). Substitution rates often vary among lineages: generation time, spontaneous mutation rate, and fixation probabilities of new mutations could all contribute to the variation of substitution rates (*Ohta, 1993*; *Lynch, 2010*). Recent evidence suggests mutation rates are even variable among human populations (*Harris, 2015*; *DeWitt et al., 2021*). As such variation affects how fast the molecular clock ticks, reconstructing gene genealogies among different species sometimes accounts for species-specific rates of evolution (*Lepage et al., 2007*). However, under the standard coalescent framework, empirical studies of within- and between-species variation tend to ignore rate variation among populations (*Costa and Wilkinson-Herbots, 2017*; *Wolf and Ellegren, 2017*;

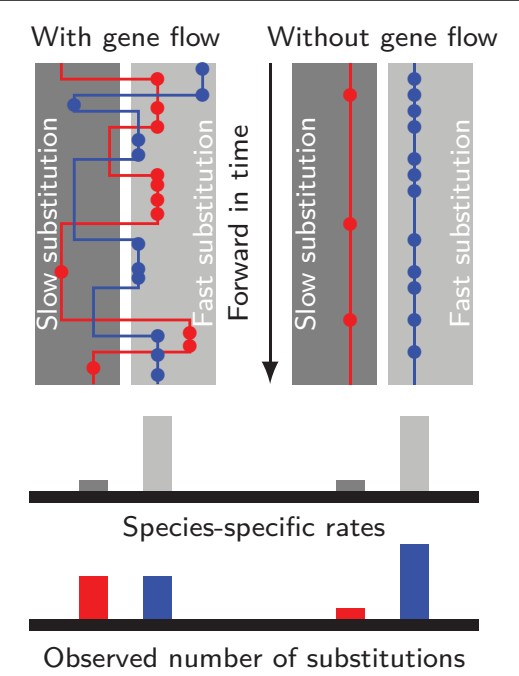

**Figure 1.** Gene flow interacts with divergent substitution rates and affects observed numbers of substitutions. Each gray block represents a species with its species-specific substitution rate. Solid lines represent gene genealogies prior to coalescence, and horizontal jumps between species represent interspecific gene flow. Dots are substitutions. When a gene sequence inherits mutations derived under multiple rates, the number of substitutions it carries will reflect a mixture of substitution rates among different species. If gene flow is strong, each lineage carries a similar number of substitutions; If gene flow is weak, genes evolve independently with species-specific rates, and the distribution of substitutions in each lineage will likely be skewed towards the distribution of species-specific substitution rates.

*Kautt et al., 2020*). The latter is partly based on a popular assumption in coalescent theory that neutral mutation rate is constant for a given locus across the whole genealogy (*Hudson, 1990*). Hybridization and speciation lie in the gray zone of these extremes, and have their own problem: molecular clocks from different lineages could be mixed by cross-species gene flow — a gene could evolve under one clock before it flows into another species and switches to evolve according to a different clock (*Figure 1*). This mixing process is largely outside the scope of existing theories and has received little attention from empirical studies.

However, if mixing between molecular clocks exists and can be measured, its strength could carry information about gene flow, which is important for studying reproductive isolation between incipient species. Genomic regions responsible for reproductive isolation lead to locally elevated genomic divergence ("genomic islands"), often caused by linked genomic regions experiencing less gene flow ("barrier loci"; *Nosil et al., 2009*; *Michel et al., 2010*; *Renaut et al., 2013*; *Payseur and Rieseberg, 2016*). In studying a hybrid zone between two butterfly species, *Papilio syfanius* and *Papilio maackii*, we discovered evidence for unequal genome-wide substitution rates between the two species. Using this system, we investigate the interaction between unequal substitution rates and gene flow, and whether this interaction reveals new information on reproductive isolation.

As these butterflies are rare species, occurring in a remote region of China, and are hard to collect, we employed methods based on analysis of whole-genome sequences of a few specimens. We hope that these methods may prove of use in studying other rare or perhaps endangered species where few individuals can be sacrificed. Results will follow two parallel lines: first, we provide evidence that genomic islands are associated with barrier loci. Then we infer the existence of unequal substitution rates. Finally, using a coalescent model, we calculate the relationship between the magnitude of genomic islands and the degree of mixing between substitution rates in linked regions. Throughout the analysis, we assume reverse mutations are rare, so that higher substitution rates always lead to higher numbers of observed substitutions.

## Results
### Divergent sister species with ongoing hybridization

We sampled 11 males of *P. syfanius* and *P. maackii* across a geographic transect (*Figure 2A*, dashed lines) covering both pure populations (in the sense of being geographically far from the hybrid zone) and hybrid populations. We also include four outgroup species, two of which have chromosome-level genome assemblies (*P. bianor*, *P. xuthus*; *Lu et al., 2019*; *Li et al., 2015*), while the other two (*P. arcturus*, *P. dialis*) are new to this study. All samples were re-sequenced to at least 20× coverage across

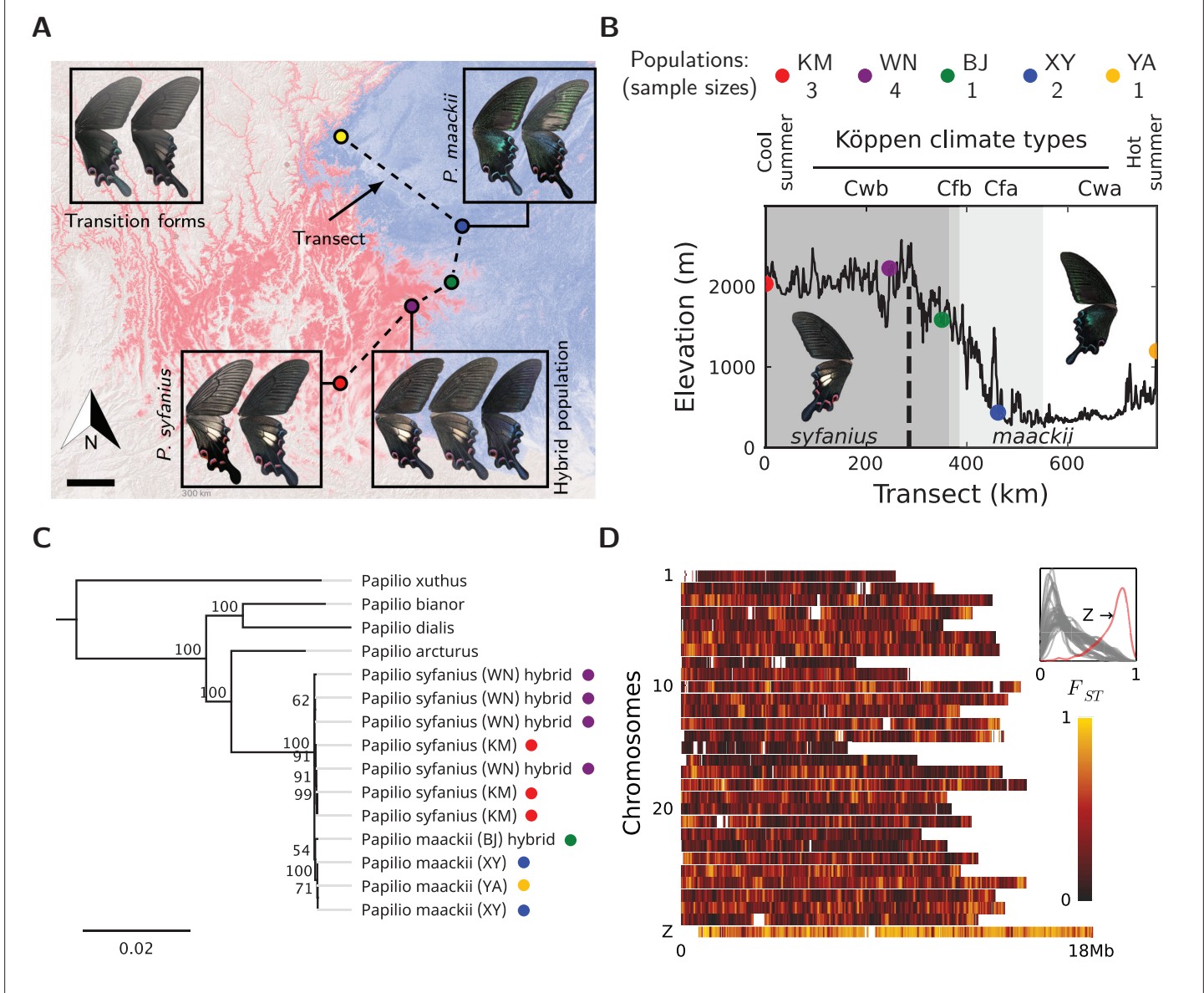

**Figure 2.** Overview of the study system. (**A**) The geographic distribution of *P. syfanius* (red) and *P. maackii* (blue). Scale-bar: 100 km. Dashed line is the sampling transect covering five populations (colored circles). (**B**) Elevation, climate, and sample sizes along the transect. (**C**) Mitochondrial tree with four outgroups. (**D**)$F_{ST}$ across chromosomes (50 kb windows with 10 kb increments). The inset shows the estimated density of $F_{ST}$ on each chromosome.

The online version of this article includes the following source data and figure supplement(s) for figure 2:

**Source data 1.** Estimated $F_{ST}$ between populations KM and XY (50 kb windows with 10 kb increments).

**Source data 2.** $D_{XY}$ and $\pi$ estimated for populations KM and XY (non-overlapping 10 kb windows).

**Source data 3.** The phylogeny and alignment among mitochondrial genomes.

**Source data 4.** Results and data for MaxEnt species distribution models.

**Source data 5.** Sample information.

**Figure supplement 1.** Polymorphic mimetic color patterns in *P. syfanius*.

**Figure supplement 2.** Estimated seasonal occurrence times of *P. syfanius* and *P. maackii* adults.

**Figure supplement 3.** Estimated geographic distribution of *P. syfanius* and *P. maackii ssp. han*.

**Figure supplement 4.** Window-based $D_{XY}$ and $\pi$ estimated for populations KM and XY.

the genome and mapped to the genome assembly of *P. bianor*. Among sampled local populations, *P. syfanius* inhabits the highlands of Southwest China (*Figure 2A*, red region), whereas *P. maackii* dominates at lower elevations (*Figure 2A*, blue region) (see *Figure 2—figure supplement 3* and *Figure 2—source data 4* for the complete distribution). The two lineages form a spatially contiguous hybrid zone at the edge of the Hengduan Mountains (*Figure 2B*) with individuals exhibiting intermediate wing patterns (*Figure 2A*: purple dot, corresponding to population WN in *Figure 2B*). Consistent with previous results (*Condamine et al., 2013*), assembled whole mitochondrial genomes are not distinct between the two lineages (*Figure 2C*, *Figure 2—source data 3*), suggesting either that divergence was recent, or that gene flow has homogenized the mitochondrial genomes. However, the two species are likely adapted to different environments associated with altitude, as several ecological traits are strongly divergent (*Kashiwabara, 1991*; *Figure 2—figure supplement 1*). Similarly, between pure populations (KM and XY in *Figure 2B*), relative divergence ($F_{ST}$) is also high across the entire nuclear genome (*Figure 2D*, *Figure 2—source data 1*). The $F_{ST}$ on autosomes averages between 0.2–0.4, and on the sex chromosome (Z-chromosome) it reaches 0.78. A highly heterogeneous landscape of $F_{ST}$ is accompanied by numerous islands of elevated sequence divergence ($D_{XY}$) and reduced genetic diversity ($\pi$) scattered across the genome (*Figure 2—figure supplement 4*, *Figure 2—source data 2*). Since animal mitochondrial DNA generally has higher mutation rates than the nuclear genome (*Haag-Liautard et al., 2008*), its low divergence between the two species are likely the result of gene flow. Overall, despite ongoing hybridization, genomes of the pure populations of *P. syfanius* and *P. maackii* are strongly differentiated.

## Genomic islands are associated with barrier loci

A natural question is whether genomic differentiation is associated with barrier loci and reproductive isolation. In other words, can $F_{ST}$ variation be attributed to gene flow variation between sister species? We suspect that barrier loci likely exist, because sequence variation between pure populations suggests that elevated $F_{ST}$ is associated with reduced $\pi$ and elevated $D_{XY}$ across autosomes (*Figure 3A*), as expected for hybridizing species (*Irwin et al., 2018*). The Z chromosome (sex chromosome) has the highest $D_{XY}$ and the lowest $\pi$ among all chromosomes (*Figure 2—figure supplement 4*), another characteristic of hybridizing species with barriers to gene flow (*Kronforst et al., 2013*).

To test for the presence of barrier loci, we augment the analysis with the sequences of four individuals from the population closest to the center of the hybrid zone (Population WN). We investigate whether differences in ancestry variation provide additional evidence for barrier loci in this hybrid zone. The underlying logic is that barrier loci will simultaneously:

1. Reduce linked $\pi$ in pure populations (*Ravinet et al., 2017*);
2. Elevate linked $D_{XY}$ between pure populations (*Ravinet et al., 2017*);
3. Elevate pairwise linkage disequilibrium in hybrid zones (*Barton, 1983*);
4. Enrich linked ancestry from one lineage in hybrid zones (*Sedghifar et al., 2016*).

Effects 3 and 4 can be bundled together as "reduced ancestry randomness" around barrier loci because both are expected if intermixing of segments of different ancestries within a genomic interval is prevented. For effects 3 and 4, because of small sample sizes, estimating site-specific statistics such as pairwise linkage disequilibrium is untenable. However, our high-quality chromosome-level reference genome enabled accurate estimation of local ancestry. As a remedy for small sample sizes, we employ two entropy metrics borrowed from signal-processing theory to quantify ancestry randomness in local regions along chromosomes (see Materials and methods). By dividing chromosomes into segments, we can extract indirect information about effects 3 and 4 at the expense of reduced genomic resolution. The proposed metrics, $S_b$ and $S_w$, correspond to the randomness of ancestry between and within individual diploid genomes from a local population (*Figure 3B*). For a cohort of ideal chromosomes with uniform recombination and marker density, if ancestry is independent between homologous chromosomes, both $S_b$ and $S_w$ increase with reduced local ancestry correlation and more balanced parental contribution (*Figure 3C*).

For a given autosomal segment, we then calculate $\pi$, $D_{XY}$, and $F_{ST}$ between pure populations, as well as entropy metrics $S_b$ and $S_w$ in hybrid individuals for the same segment. To investigate whether effects 1–4 are all present in our system, the joint range among entropy, $\pi$, and $D_{XY}$ is shown in *Figure 3D*, which suggests that low ancestry randomness (low entropy) is likely associated with reduced $\pi$ within species and elevated $D_{XY}$ between species. To further quantify such association, we

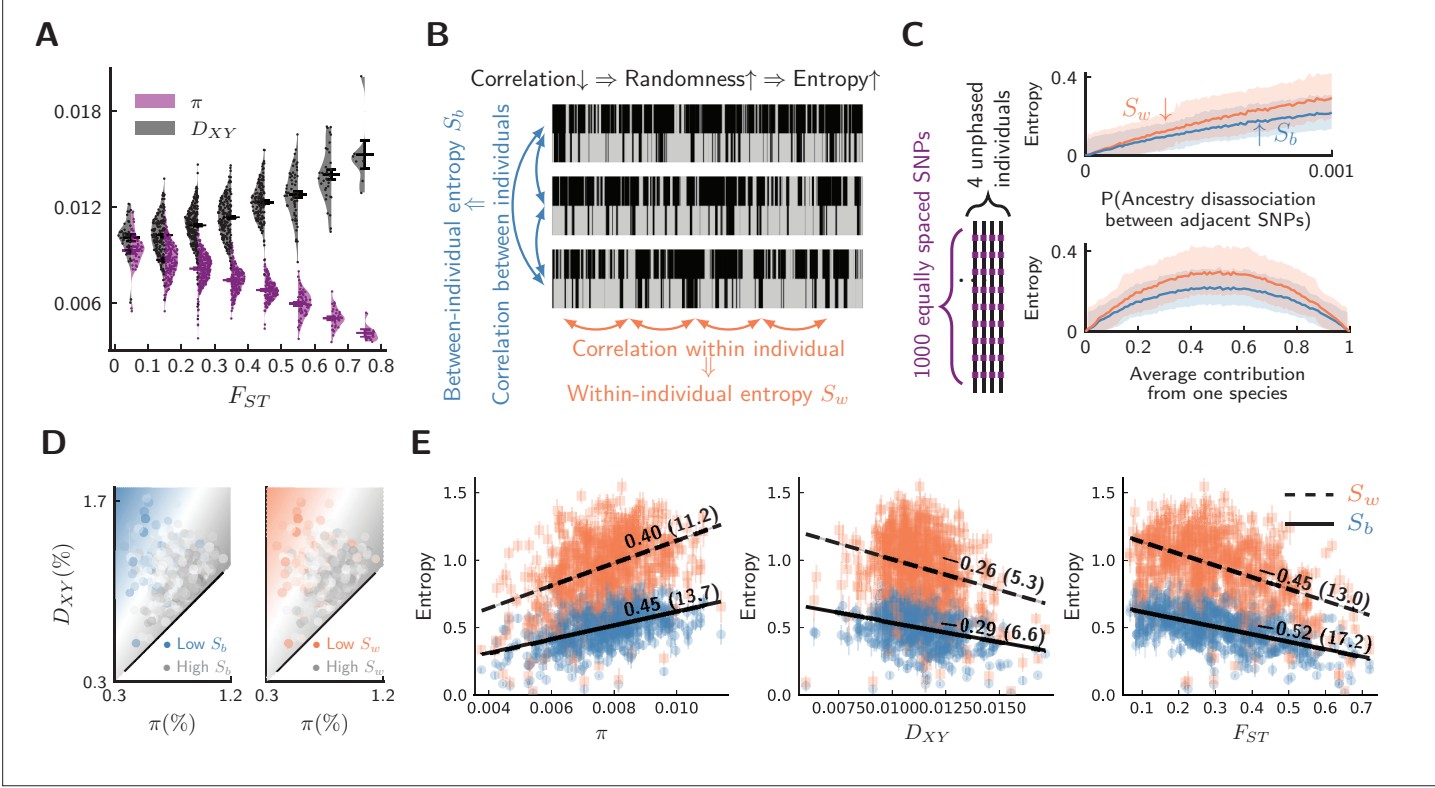

**Figure 3.** Evidence for barrier loci on autosomes. For all plots, pure populations refer to XY & KM, and hybrid population refers to WN. (**A**) Between pure populations, reduced diversity ($\pi$, showing values averaged between pure populations) is associated with increased divergence ($D_{XY}$) across autosomes (30 segments per chromosome). Error bars are standard errors. (**B**) The conceptual picture of entropy metrics on diploid, unphased ancestry signals. In a genomic window, between-individual entropy ($S_b$) measures local ancestry randomness among individuals, while within-individual entropy ($S_w$) measures ancestry randomness along a chromosomal interval. (**C**) Simulated behaviors of entropy in a simplified model of biparental local ancestry. Chromosomes are assumed to be spatially homogeneous, thus recombination rate is uniform among 1000 equally spaced SNPs, and adjacent SNPs have a single probability of ancestry disassociation. For each haplotype block with linked ancestry, its ancestry is randomly assigned according to the average contribution from each species. Each pair of haploid chromosomes are combined into an unphased ancestry signal before calculating entropy. The top plot assumes equal contribution from both species, and the bottom plot assumes ancestry disassociation probability = 0.001. Solid lines are average entropies across 1000 repeated simulations, and shaded areas represent averaged upper and lower deviations from the mean. (**D**) The joint range among entropy, $\pi$, and $D_{XY}$ across autosomes (20 segments per chromosome). Color range is normalized by the range of entropy in each plot. Gray represents higher entropy, and colored regions are associated with lower entropy. Heatmaps represent linear fits to the ensembles of points. (**E**) The correlation $\rho$ on autosomes between entropy in hybrid populations and {$\pi$, $D_{XY}$, $F_{ST}$} within and between pure populations. $\rho$ is shown above each regression line. Error bars are standard errors of entropy from 50 repeated estimates of local ancestry using software ELAI (parameters are in Materials and methods). The significance of $\rho$ is estimated using block-jackknifing among all segments: $Z$-scores are shown in parentheses.

The online version of this article includes the following source data and figure supplement(s) for figure 3:

**Source data 1.** Estimated entropy ($S_w$, $S_b$), $D_{XY}$, $\pi$, $F_{ST}$ on 20 segments per chromosome (including the Z chromosome).

**Figure supplement 1.** An example run of the local ancestry estimation software ELAI on chromosomes 1–30.

estimated Pearson's correlation coefficients ($\rho$) between entropy and the latter statistics (**Figure 3E**). These associations are strongly significant ($Z$-scores > 3). Consequently, reduced ancestry randomness in hybrids (effects 3 & 4) coincides with classical patterns of barrier loci between pure populations (effects 1 and 2). This analysis is not sufficient to exclude all alternative hypotheses. For instance, we cannot entirely exclude the possibility that patterns are driven by low-recombination regions ($S_w, S_b \downarrow$) that experience linked selection ($\pi \downarrow$) also have elevated mutation rates ($D_{XY} \uparrow$). Nonetheless, this alternative seems most unlikely, as low-recombination regions should typically be less rather than more mutable (*Lercher and Hurst, 2002*; *Jensen-Seaman et al., 2004*; *Yang et al., 2015*; *Arbeithuber et al., 2015*; *Liu et al., 2017*). Overall, adding information from hybrid populations strengthens the evidence for barrier loci acting across autosomes.

The Z chromosome was excluded from this analysis as it likely differs in mutation rate or effective population size (**Presgraves, 2018**), but its ancestry in hybrid individuals either retains purity or resembles very recent hybridization (long blocks of heterozygous ancestry, **Figure 3—figure supplement 1**). The Z chromosome has low ancestry randomness, and it also has the highest level of divergence (**Figure 2D**), both of which suggest strong barriers to gene flow between *P. syfanius* and *P. maackii* on this chromosome.

## Asymmetric site patterns

A hint that substitution rates differ between the two species comes from site-pattern asymmetry (but this asymmetry alone is insufficient to establish the existence of divergent rates). We focus on two kinds of biallelic site patterns. In the first kind, choose three taxa ($P_1$,$P_2$,$O_1$), with $P_1$=*syfanius*, $P_2$=*maackii*, while $O_1$ is an outgroup. Assuming no other factors, if substitution rates are equal between $P_1$ and $P_2$, then site patterns ($P_1$,$P_2$,$O_1$)=(A,B,B) and (B,A,B) occur with equal frequencies, where "A" and "B" represent distinct alleles. This leads to a $D$ statistic describing the asymmetry between three-taxon site patterns:

$$D_3 = D_{P_1,P_2,O_1} = \frac{n_{ABB} - n_{BAB}}{n_{ABB} + n_{BAB}},$$
(1)

where $n$ is the count of a particular site pattern across designated genomic regions. A significantly non-zero $D_3$ indicates strongly asymmetric distributions of alleles between $P_1$ and $P_2$.

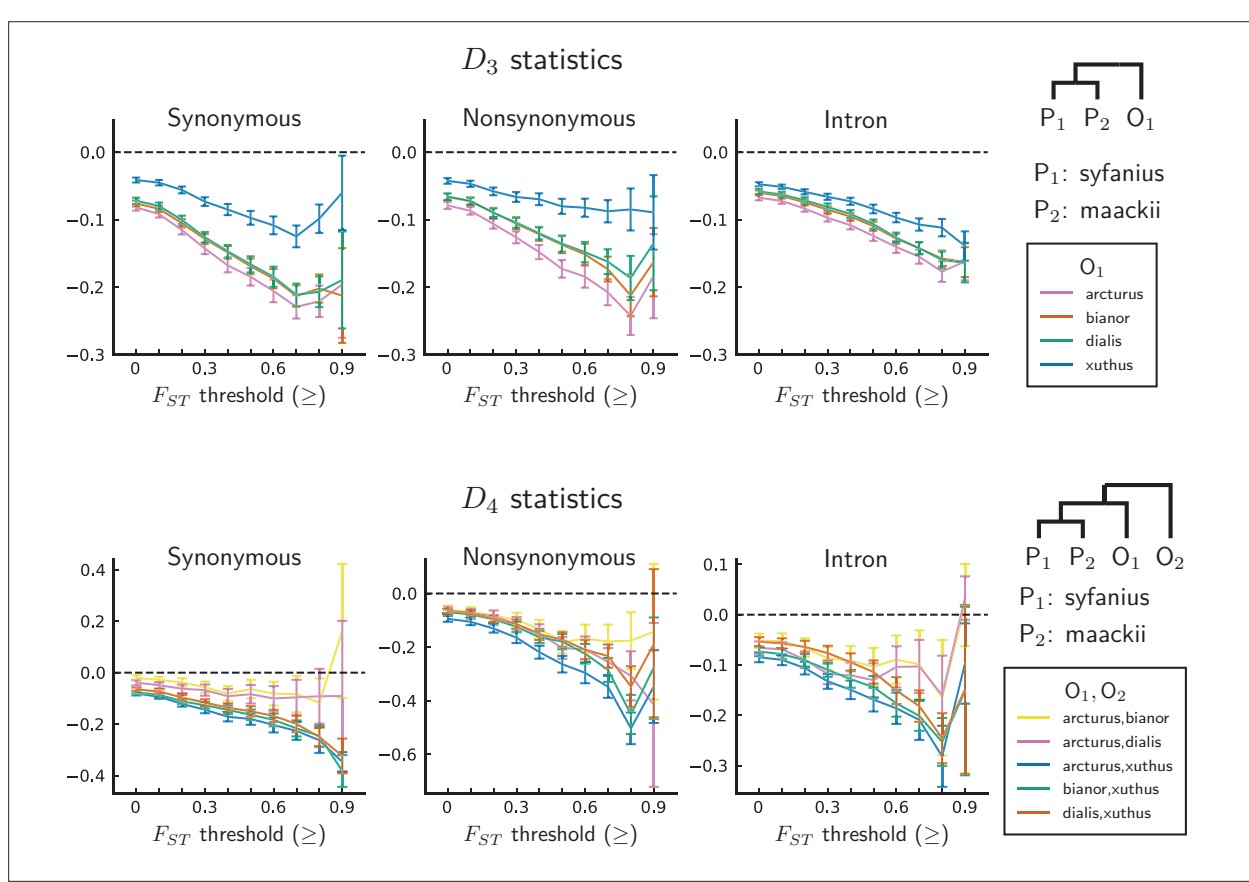

**Figure 4.** $D$ statistics are unanimously negative. For each data point, we choose an $F_{ST}$ threshold (x-axis) and report $D$ statistics on SNPs with a background $F_{ST}$ (50 kb windows and 10 kb increments) no less than the given threshold. Error bars are standard errors estimated using block-jackknife with 1 Mb blocks. The aberrant behavior for the highest $F_{ST}$ bins is we believe mostly due to low sample sizes for these bins.

The online version of this article includes the following source data and figure supplement(s) for figure 4:

**Source data 1.** $D_3$, $D_4$ statistics with their $Z$-scores.

**Figure supplement 1.** $D_3$, $D_4$ statistics plotted with their $Z$-scores.

**Table 1.** Coverage abnormality in SNPs from coding sequences (CDSs).
Abnormal coverage is inferred if the average coverage of a CDS exceeds twice of the median coverage of all CDSs in the genome.

| | |
|---|---|
| Number of SNPs from CDSs with abnormal coverage | 1993 |
| Total number of SNPs from all CDSs | 1060525 |

In a second kind of site pattern test, we compare four taxa ($P_1, P_2, O_1, O_2$) and calculate a similar statistic between site patterns (A,B,B,A) vs (B,A,B,A):

$$D_4 = D_{P_1,P_2,O_1,O_2} = \frac{n_{ABBA} - n_{BABA}}{n_{ABBA} + n_{BABA}} \tag{2}$$

Classically, a significantly non-zero $D_4$ suggests that gene flow occurs between an outgroup and either $P_1$ or $P_2$, thus it is widely used to detect hybridization (the ABBA-BABA test) (***Durand et al., 2011***; ***Hibbins and Hahn, 2022***). Here, $D_4$ is used more generally as an additional metric of site pattern asymmetry.

We compute both $D_3$ and $D_4$ on synonymous, nonsynonymous, and intronic sites, with *P. syfanius* and *P. maackii* samples taken from pure populations (KM and XY). For each type of site, we progressively exclude regions with local $F_{ST}$ below a certain threshold, and report $D$ statistics on the remaining sites in order to show the increasing site-pattern asymmetry in more divergent regions (***Figure 4***). $D_3$ is significantly negative regardless of outgroup or site type for most $F_{ST}$ thresholds, and $D_4$ is also significantly negative for most outgroup combinations when computed across the entire genome (Z-scores are shown in ***Figure 4—figure supplement 1***), proving that site-patterns are strongly asymmetric between *P. syfanius* and *P. maackii*. Importantly, the direction of asymmetry is nearly identical across all outgroup comparisons. This asymmetry cannot be attributed to batch-specific variation as all samples were processed and sequenced in a single run, and variants were always called on all individuals of *P. syfanius* and *P. maackii*. Sequencing coverage is normal for most annotated genes used in the analysis (***Table 1***), suggesting that asymmetry is not due to systematic copy-number variation that could affect variant calls. Nonetheless, two independent processes of evolution could explain observed asymmetric site patterns. In hypothesis I (***Figure 5A***, left), asymmetry is generated via stronger gene flow between *P. syfanius* and outgroups, leading to biased allele-sharing. In hypothesis II, site pattern asymmetry is due to unequal substitution rates between *P. syfanius* and *P. maackii*, which is further modified by recurrent mutations in all four outgroups (***Figure 5A***, right). We test each hypothesis below.

## Hybridization with outgroups does not explain site-pattern asymmetry

We first test hypothesis I by phylogenetic reconstruction using SNPs in annotated regions. We construct local gene trees for each 50 kb non-overlapping window for all samples, including the four outgroup species. As biased gene flow with outgroups should rupture the monophyletic relationship among all *P. syfanius* +*P. maackii* individuals, the fraction of windows producing paraphyletic gene trees can be used to assess the potential impact of gene flow. However, almost all gene trees show the expected monophyletic relationship (***Figure 5B***). This conclusion is independent of the level of support used to filter out genomic windows with ambiguous topologies (see Materials and methods). Consequently, hardly any window shows a phylogenetic signal of hybridization with outgroups. Nonetheless, branch lengths in reconstructed gene trees suggest possible substitution rate divergence between the two species: Among highly supported monophyletic trees (bootstrap support ≥ 95), *P. maackii* (in populations YA, BJ, XY) is always significantly more distant than *P. syfanius* from the most recent common ancestor of *P. maackii* +*P. syfanius* (***Figure 5C***, significance levels reported via a Wilcoxon signed-rank test for each pair of individuals, see ***Figure 5—source data 2***). Second, the direction of allele sharing in the $D$ statistics is unanimously biased towards *P. syfanius*. If hypothesis I were true, hybridization with outgroups is required to take place mainly in the highland lineage. There is no evidence to support why the highland lineage should receive more gene flow based on current geographic distributions, as outgroups *P. xuthus* and *P. bianor* overlap broadly with both *P. maackii* and *P. syfanius*, while outgroup *P. dialis* is sympatric only with *P. maackii* (***Condamine et al., 2013***). Although it is possible that historical and modern geographic distributions differ, and archaic gene flow might have occurred

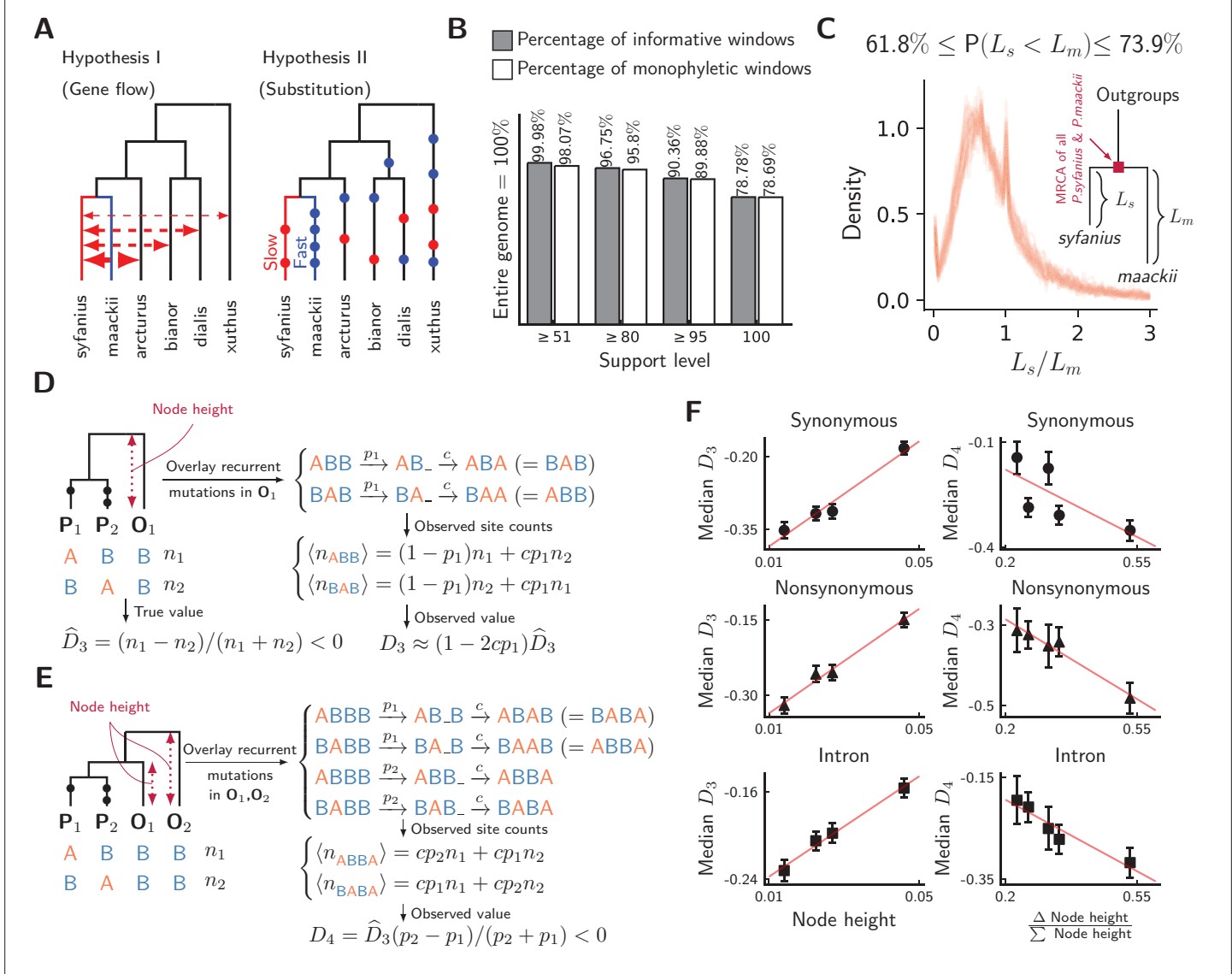

**Figure 5.** Unequal substitution rates between pure population of *P. syfanius* and *P. maackii*. (**A**) Two hypotheses to explain negative $D$ statistics. (**B**) The percentage of local gene trees (50 kb non-overlapping windows) where *P. syfanius* and *P. maackii* are together monophyletic. For each level of support, we filter out windows with *Support*(monophyly) and *Support*(paraphyly) below a given level, and report both the percentage of windows passing the filter (informative windows) and the percentage of monophyletic windows. (**C**) The distribution of *P. syfanius* branch lengths ($L_s$) relative to those of *P. maackii* ($L_m$) among highly supported monophyletic trees (*Support* ≥95). Each curve corresponds to a pairwise comparison between a *syfanius* individual and a *maackii* individual. Branch lengths are distances from tips to the most-recent common ancestor of all *syfanius* +*maackii* individuals. (**D**) Behavior of $D_3$ under divergent substitution rates and recurrent mutations in outgroups ($O_1$). (**E**) Behavior of $D_4$ under divergent substitution rates and recurrent mutations in outgroups ($O_1$ & $O_2$). (**F**) Left: Median $D_3$ is positively correlated with node height; Right: Median $D_4$ is negatively correlated with ($\Delta$Node height)/($\Sigma$Node height). Node height is used as a proxy for the probability of recurrent mutation ($p_i$).

The online version of this article includes the following source data for figure 5:

**Source data 1.** Concatenated gene trees based on 50 kb windows and the underlying support for each binary split.

**Source data 2.** Results of the signed-rank test on branch elongation in *P. maackii*.

---

preferentially with *P. syfanius*, it should still leave some phylogenetic signal of introgression. Overall, we find little evidence for biased hybridization required by hypothesis I.

One might worry that by rejecting hypothesis I, we also throw doubt on widely accepted conclusions of the ABBA-BABA test for gene flow in other systems (that a significantly nonzero $D_4$ implies hybridization with outgroups) (***Durand et al., 2011***). However, in the next section, we show why $D_3$

and $D_4$ are fully consistent with hypothesis II, and so this is just a special case where the ABBA-BABA test produces a false positive for gene flow.

## Evidence for divergent substitution rates

In hypothesis II, divergent substitution rates between *P. maackii* and *P. syfanius* interact with recurrent mutations in outgroups to produce asymmetric site patterns. To understand its effect on $D$ statistics, consider a simplified model of recurrent mutation (*Figure 5D,E*), where a site in outgroup   mutates with probability $p_i$, producing the same derived allele with probability $c$. When averaged across the genome, $c$ can be treated as a constant, and $p_i$ increases with distance to the outgroup. In the absence of gene flow, for three-taxon patterns, recurrent mutations modify $D_3$ by a factor of approximately $(1 - 2cp_1)$ (*Figure 5D*, see Materials and methods), and observed $D_3$ will thus be positively correlated with $p_1$. For four-taxon patterns, it can be shown that observed $D_4$ is always negative due to larger probabilities of recurrent mutation in more distant outgroups (*Figure 5E*, see Materials and methods). Assuming no significant contribution of incomplete lineage sorting (*Maddison and Knowles, 2006*), the value of $D_4$ becomes more negative with increasing $\Delta p_i / \Sigma p_i = (p_2 - p_1)/(p_2 + p_1)$.

To test these signatures, we employ estimated node heights of outgroups in the mitochondrial tree (*Figure 2C*) as proxies for outgroup distance, and hence for the relative probability of recurrent mutation ($\rho_i$). In line with expected signatures, we find that observed $D_3$ indeed increases with node height (*Figure 5F*, left), and observed $D_4$ decreases with ($\Delta$ Node height/$\Sigma$ Node height; *Figure 5F*, right). Thus, the directions and magnitudes of both $D$ statistics are congruent with hypothesis II. As hypothesis II naturally predicts unanimously negative $D_3$ and $D_4$ as well as their relative magnitudes among different outgroup combinations, it is more parsimonious than hypothesis I. Hence, divergent substitution rates likely exist between *P. syfanius* and *P. maackii*.

## Rate-mixing at migration-drift equilibrium

Having tested for the existence of divergent substitution rates, we now explore how they become mixed by gene flow using a coalescent framework. In particular, we will assess whether the conceptual picture in *Figure 1* can be recovered quantitatively in the butterfly system. As gene flow is ongoing between the two lineages, consider two haploid populations of size $N$ exchanging genes at rate $m$ (*Figure 6A*). This simple isolation-with-migration (IM) model at equilibrium has relative divergence (*Notohara, 1990*)

$$F_{ST} = \frac{1}{1 + 4Nm} \tag{3}$$

To quantify the signature of mixed rates, consider the asymmetry in observed numbers of substitutions (circles in *Figure 6A*). Take a pair of sequences from two populations. Let $\langle n_1 \rangle$ be the expected number of derived alleles exclusive to the sequence in population 1, and let $\langle n_2 \rangle$ be the same expected number in population 2. Their ratio is defined as observed rate ratio:

$$r = \frac{\langle n_2 \rangle}{\langle n_1 \rangle} \tag{4}$$

Evidently, $r = 1$ is the symmetric point where both sequences have the same number of derived alleles. Further, let the actual substitution rate in population 1 be $\mu_1$, and the actual substitution rate in population 2 be $\mu_2$. The ratio between the two actual rates are defined as the true rate ratio:

$$r_0 = \frac{\mu_2}{\mu_1} \tag{5}$$

At migration-drift equilibrium, observed rate ratio ($r$) and observed divergence ($F_{ST}$) are related by the following formula parameterized singly by $r_0$ (*Figure 6B*, see Materials and methods):

$$r = \frac{1 + r_0 + F_{ST}(r_0 - 1)}{1 + r_0 - F_{ST}(r_0 - 1)}, \tag{6}$$

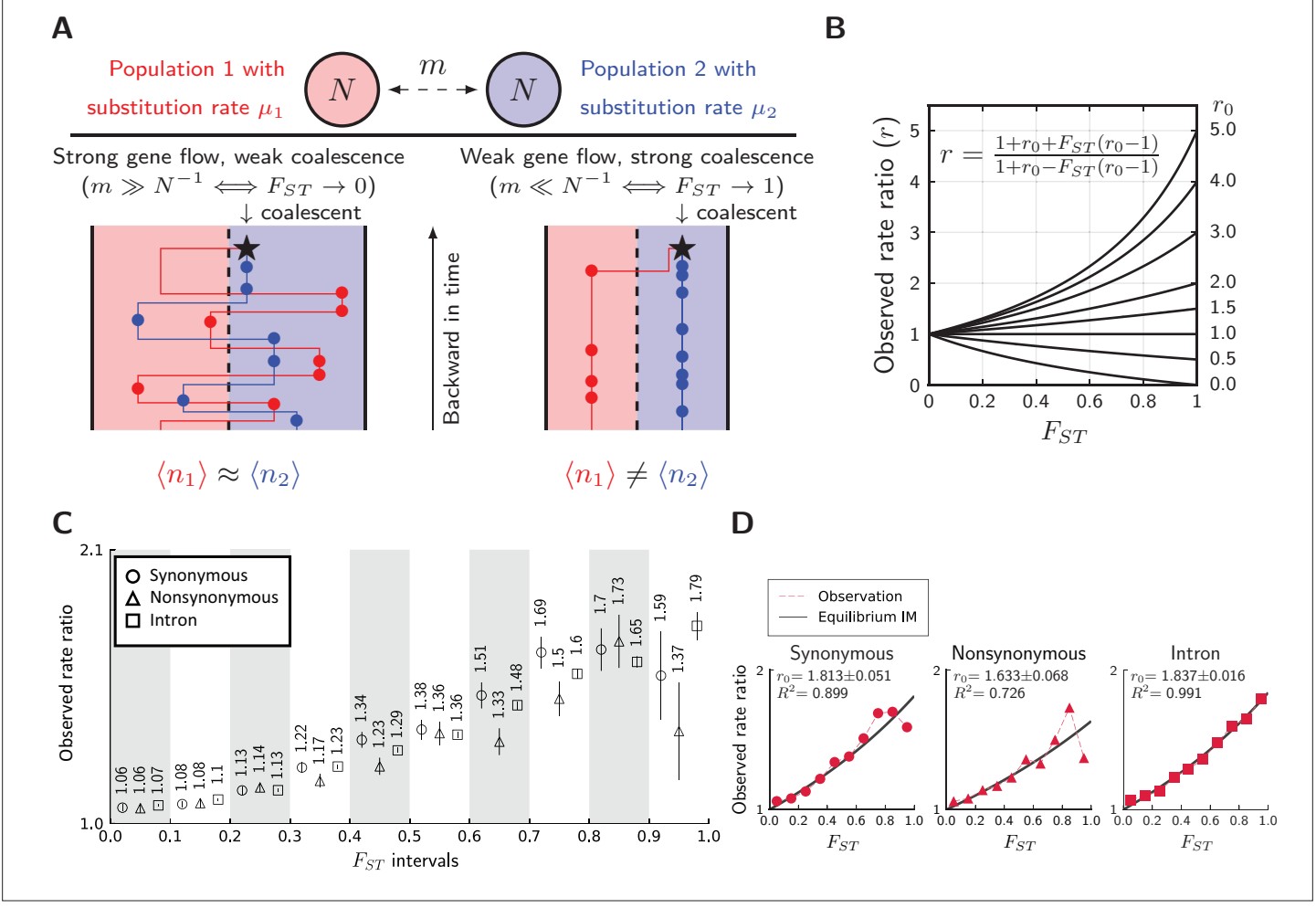

$\rho_0$

**Figure 6.** Divergence is correlated with increased differences in the relative number of substitutions. (**A**) Behavior of the equilibrium isolation-with-migration model with divergent substitution rates. If coalescence is weaker than gene flow, each lineage has a similar number of derived alleles. If coalescence is stronger than gene flow, lineages sampled from the population with a faster substitution rate will also inherit more derived alleles. (**B**) Theoretical relationship between observed rate ratio ($r$) and relative divergence ($F_{ST}$), parameterized by the true ratio $r_0$ of substitution rates. (**C**) Observed rate ratios between *P. maackii* (population XY) and *P. syfanius* (Population KM), partitioned by ten $F_{ST}$ intervals. Error bars are standard errors calculated using 1 Mb block-jackknifing. (**D**) The theoretical relationship between $r$ and $F_{ST}$ is a good fit to observation.

The online version of this article includes the following figure supplement(s) for figure 6:

**Figure supplement 1.** Simulated results from equilibrium symmetric migration models of rate-mixing.

**Figure supplement 2.** Simulated results from equilibrium conservative migration models of rate-mixing.

**Figure supplement 3.** Simulated results from equilibrium stepping-stone models of rate-mixing.

which translates into

$$F_{ST} \approx \frac{r-1}{r_0-1} \quad \text{if } r_0 \approx 1 \tag{7}$$

These formulae indicate that unequal substitution rates are more mixed in regions with lower genomic divergence. *Equation 7* is surprising, because it reveals that the remaining fraction of substitution rate divergence (($r-1$)/($r_0-1$)) is almost the same as $F_{ST}$, which corresponds to the fraction of genetic variance explained by population structure (**Wright, 1949**). In *Figure 6B*, the relationship between $r$ and $F_{ST}$ is still largely linear when one species evolves three times as fast as its sister species ($r_0 = 3$), and so *Equation 7* might be robust under biologically realistic rates of substitutions between

incipient species. Using extensive simulations (*Figure 6—figure supplement 1* to *Figure 6—figure supplement 3*), we show that the full formula (*Equation 6*) is also robust in a number of equilibrium population structures.

To test whether such predictions are met, we calculate $r$ on synonymous, nonsynonymous, and intronic sites partitioned by their local $F_{ST}$ values between pure populations (KM & XY), and recover a similar monotonic relationship across most $F_{ST}$ partitions (*Figure 6C*). For introns, the observed relationship between $r$ and $F_{ST}$ is a near perfect fit to *Equation 6* (*Figure 6D*, squares), with an estimated $r_0$ = 1.837. For synonymous (*Figure 6D*, circles) and nonsynonymous sites (*Figure 6D*, triangles), *Equation 6* also provides an excellent fit in regions with low to intermediate $F_{ST}$. Estimated $r_0$ for synonymous sites is 1.813, close to that of introns, while nonsynonymous sites have a considerably lower $r_0$ of 1.633. If introns and synonymous substitutions are approximately neutral, we infer that neutral substitution rates are about 80% greater in the lowland species.

## Discussion

### Entropy as a useful measure of ancestry randomness

A critical step in our analysis is to associate genomic islands with barrier loci. Conventionally, information on increased association between barrier sites in hybrid populations comes from two-locus linkage disequilibrium (*Slatkin, 1975*). Empirical studies on hybrid populations frequently use such statistics to strengthen the evidence for barriers to gene flow (*Knief et al., 2019*; *Wang et al., 2022*). Alternatively, if phased haplotypes can be sequenced, the length of ancestry tracts (*Sedghifar et al., 2016*) or the density of ancestry junctions (*Janzen et al., 2018*; *Wang et al., 2022*) also carry information on barrier loci. All these methods break down with small numbers of unphased samples, as forced phasing can produce a large number of false ancestry junctions. However, conventional notions such as "ancestry tract length" are not definable for unphased local ancestry. The dilemma forces us to consider more robust statistics that carry information on ancestry association even in unphased data.

The development of both entropy metrics follows three intuitions:

- Vectorization: ancestry is a categorical concept, and it should map to a signal containing the contribution of each source population.
- Conservation law: the total ancestry from all sources at a particular locus is conserved.
- Highly (auto)correlated signals have a concentrated spectrum (low spectral entropy).

Since entropy carries information about the degree of ancestry correlation, it will decrease in regions of low recombination and high genetic relatedness. If genetic relatedness is produced by inbreeding, it should affect the entire genome in a similar way, and between-individual entropy ($S_b$) will be similar across different parts of the genome. However, if inbreeding is so severe that $S_b$ is globally zero, it will not be an informative metric. It is worth noting that within-individual entropy $S_w$ shares a conceptual similarity to the wavelet transform of ancestry signals (*Pugach et al., 2011*; *Sanderson et al., 2015*). These entropy metrics will be particularly useful for window-based genomic analysis of ancestry correlation with limited sampling, and they are also compatible with larger sample sizes and more than two parental species. (The formulae of entropy with two parental species are presented in Materials and methods, and mathematical details are discussed in Appendix 1.) Nonetheless, this entropy approach assumes local ancestry can be accurately inferred, which will be challenging for studies with low-coverage sequencing, non-chromosomal genome assembly, or lacking reference populations.

### Potential mechanisms of divergent substitution rates

Interestingly, a higher substitution rate in the lowland lineage *P. maackii* is congruent with the evolutionary speed hypothesis (*Rensch, 1959*), where evolution accelerates in warmer climates. Our finding echoes the results of many previous studies (*Gillooly et al., 2005*; *Wright et al., 2006*; *Lin et al., 2019*; *Ivan et al., 2022*). Without measuring the per-generation mutation rates in both species, it is unclear what mechanisms cause increased substitution rates, but the lowland lineage typically has an additional autumn brood that is absent in *P. syfanius* (*Takasaki et al., 2007*; *Figure 2—figure supplement 2*). Warmer temperatures in lowland habitats might also increase spontaneous mutation rates in ectothermic insects (*Waldvogel and Pfenninger, 2021*). Both mechanisms could produce increased substitution rates in the lowland species. It is surprising that estimated $r_0$ between synonymous sites

and introns agree with each other under such a coarse framework, and estimated $r_0$ for nonsynonymous sites is considerably lower. Less asymmetric rates for nonsynonymous substitutions could perhaps be explained by the nearly neutral theory (*Ohta, 1993*), which argues that many nonsynonymous mutations are mildly deleterious, and selection is more efficient in suppressing them in larger populations and slowing down substitutions. In the field, the lowland species often appears in larger numbers with well-connected habitats, while the highland species faces a highly heterogeneous landscape of the Hengduan Mountains, which could lead to differences in effective population sizes required by our expectation under the nearly neutral theory.

## Are introgression tests robust to substitution rate variation?

An additional result from our study is that divergent substitution rates might produce spuriously non-zero $D_4$ statistics when combined with recurrent mutations, which could increase the false positive rate of the ABBA-BABA test (*Durand et al., 2011*). This phenomenon has been suspected in humans (*Amos, 2020*), and is certainly a theoretical possibility (*Hibbins and Hahn, 2022*), but has not been tested in most empirical studies.

A wide class of introgression tests targeting gene flow between outgroups and a pair of taxa are based on site pattern information. Hahn's $D_3$ computes absolute sequence divergence between groups in a triplet of species (*Hahn and Hibbins, 2019*), and will be affected by unequal substitution rates in a similar way to our $D_3$. Martin's $f_d$ computes the same numerator as our $D_4$ (*Martin et al., 2015*), so it could also produce false positives under similar situations. Related statistics include $D_p$ (*Hamlin et al., 2020*), $D_f$ (*Pfeifer and Kapan, 2019*). A general guideline for site-pattern based statistics is that the focal pair of taxa are closely related such that substitution rates do not differ, while outgroups should not be too distant to minimize recurrent mutations (*Hibbins and Hahn, 2022*). However, whether these assumptions are met in empirical studies is worthy of investigation, and our system provides a counterexample even between sister species with ongoing hybridization.

A separate pitfall might occur if introgression tests based on site-patterns are applied to genomic windows to locate regions introgressed from outgroups. In our case, $F_{ST}$ peaks have the most asymmetric substitution rates between *P. maackii* and *P. syfanius*, thus they will most likely be associated with false-positive $D$ statistics. This could lead to the incorrect interpretation that some barrier loci ("speciation genes") are introgressed from outgroups – a popular hypothesis in adaptive introgression and hybrid speciation, see *Edelman and Mallet, 2021*.

To this end, we speculate that using appropriate substitution models to infer gene tree topology will perform better in assessing the impact of introgression with outgroups.

## The conceptual picture of rate-mixing

In the gray zone of incomplete speciation, interspecific hybrids bridge between gene pools of divergent lineages (*Mallet, 2005*). We here demonstrate a similar role of hybridization in coupling and mixing differing substitution rates. Divergent rates of substitution carry information about outgroups, while divergence based on allele frequency differences does not. Preserving divergent substitution rates is a stronger effect than maintenance of allele frequency differences, because divergence of allele frequencies is a prerequisite for rate preservation. This dependency can be coarsely quantified across the genome by the relationship between observed rate ratio $r$ and relative divergence $F_{ST}$ in an equilibrium system of hybridizing populations (*Equation 6*). At migration-drift equilibrium, it is not surprising that divergent substitution rates are associated with relative divergence. In *Figure 6A*, when coalescence occurs rapidly compared to gene flow, most substitutions separating individuals are species-specific. However, when gene flow is faster than coalescence, individuals will carry substitutions that occurred in both species. This could have important implications, because preserving lineage-specific substitution rates as measured by $r$ might not require low absolute rates of gene flow. Instead, reducing effective population sizes via recurrent linked selection might achieve a similar result in populations at equilibrium ($N \downarrow \Rightarrow F_{ST} \uparrow \Rightarrow r \uparrow$).

The theory in its present form has several limitations. First, mutations follow the infinite-site model (*Kimura, 1969*), so that reverse mutations, double mutations, and substitution types are not accurately reflected in the estimated ratio between species-specific substitution rates. Second, population structure is assumed at equilibrium, whereas real data could carry footprints from non-equilibrium demographic processes (e.g. secondary contact) (*Hey, 2010*). Third, there could be considerable

population structure within each species contributing to elevated $F_{ST}$ but not necessarily to rate-divergence. This effect could be seen in simulated stepping-stone models (*Figure 6—figure supplement 3*) and will underestimate the level of rate divergence. Fourth, substitution is stochastic across the genome, and accurately estimating observed rate ratio $r$ relies on averaging substitution numbers across a large number of sites. This poses a problem if sites linked to high $F_{ST}$ regions are rare (i.e. few genomic islands). Lastly, as the theory is built upon the neutral coalescent, it is best suited for studying behaviors of neutral sites.

Nonetheless, the monotonic relationship between $r$ and $F_{ST}$ (i.e. larger sequence divergence is associated with more asymmetric substitution rates) might be qualitatively robust regardless of the aforementioned caveats. For instance, even for hybrid zones formed by recent secondary contact, reducing the absolute rate of gene flow by barrier loci in principle also keeps divergent rates from mixing, simply because it prevents substitutions accumulated in the allopatric phase from flowing between species.

In conclusion, our study characterizes several genomic consequences of the rate-mixing process when molecular clocks tick at different speeds between hybridizing lineages. This process provides new information on reproductive isolation but also leads to pitfalls in interpreting results of popular introgression tests. As this phenomenon is neglected in most studies of hybridization and speciation, its full scope awaits further investigation in both theories and empirical systems.

## Materials and methods
### Museum specimens and climate data
Museum specimens with verifiable locality data of all species were gathered from The University Museum of The University of Tokyo (*Harada et al., 2012*; *Yago et al., 2021*), Global Biodiversity Information Facility (*The Global Biodiversity Information Facility, 2021b*; *The Global Biodiversity Information Facility, 2021a*), and individual collectors (*Figure 2—source data 5*). Records of *P. maackii* from Japan, Korea and NE China were excluded from the analysis, so that most *P. maackii* individuals correspond to *ssp. han*, the subspecies that hybridizes with *P. syfanius*. Spatial principal component analysis was performed on elevation, maximum temperature of warmest month, minimum temperature of coldest month, and annual precipitation, all with 30s resolution from WorldClim-2 (*Fick and Hijmans, 2017*). The first two PCAs, combined with tree cover (*Hansen et al., 2013*), were used in MaxEnt-3.4.1 to produce species distribution models that use known localities to predict occurrence probabilities across the entire landscape (*Phillips et al., 2017*). Outputs were trimmed near known boundaries of each species. See *Figure 2—figure supplement 3* for the final result.

### Sampling, re-sequencing, and mitochondrial phylogeny
Eleven males of *P. syfanius* and *P. maackii*, with one male of *P. arcturus* and one male of *P. dialis* were collected in the field between July and August in 2018 (*Figure 2—source data 5*), and were stored in RNAlater at –20 °C prior to DNA extraction. E.Z.N.A Tissue DNA kit was used to extract genomic DNA, and KAPA DNA HyperPlus 1/4 was used for library preparation, with an insert size of 350 bp and 2 PCR cycles. The library is sequenced on a Illumina NovaSeq machine with paired-end reads of 150 bp. Adaptors were trimmed using Cutadapt-1.8.1, and subsequently the reads were mapped to the reference genome of *P. bianor* with BWA-0.7.15, then deduplicated and sorted via Picard-Tools-2.9.0. The average coverage among 13 individuals in non-repetitive regions varies between 20× and 30×. Variants were called twice using BCFtools-1.9 – the first including all samples, used in analyses involving outgroups, and the second excluding *P. arcturus* and *P. dialis*, used in all other analyses. The following thresholds were used to filter variants: $10N <$ DP $< 50N$, where $N$ is the sample size; QUAL > 30; MQ > 40; MQ0F < 0.2. As a comparison, we also called variants with GATK4 and followed its best practices, and 93% of post-filtered SNPs called by GATK4 overlapped with those called by BCFtools. We used SNPs called by BCFtools throughout the analysis. Mitochondrial genomes were assembled from trimmed reads with NOVOPlasty-4.3.1 (*Dierckxsens et al., 2017*), using a published mitochondrial ND5 gene sequence of *P. maackii* as a bait (NCBI accession number: AB239823.1). We also used the following published mitochondrial genomes (NCBI accession numbers): KR822739.1 (*Papilio glaucus*), NC_029244.1 (*Papilio xuthus*), JN019809.1 (*Papilio bianor*). The neighbor-joining mitochondrial phylogeny was built with Geneious Prime-2021.2.2 (genetic distance model: Tamura-Nei), and

we used $10^4$ replicates for bootstrapping. The reference genome of *P. xuthus* was previously aligned to the genome of *P. bianor* and we used this alignment directly in all analysis (*Lu et al., 2019*).

## Calculating site-pattern asymmetry

Given a species tree {{P$_1$,P$_2$},O}, where P$_1$ and P$_2$ are sister species and O is an outgroup, if mutation rates are equal between {P$_1$,P$_2$}, and no gene flow with O, then on average the number of derived alleles in P$_1$ should equal the number of derived alleles in P$_2$. Let $\mathcal{S}$ be a collection of sites, $f_s$ be the frequency of a particular site pattern at site $s \in \mathcal{S}$. "ABB" be the pattern where only P$_2$ and O share the same allele, and "BAB" be the pattern where only P$_1$ and O share the same allele, then the three-species $D_3$ statistic is calculated as

$$D_{\text{P}_1,\text{P}_2,\text{O}} = \frac{\sum_{s\in\mathcal{S}}(f_{s,\text{ABB}} - f_{s,\text{BAB}})}{\sum_{s\in\mathcal{S}}(f_{s,\text{ABB}} + f_{s,\text{BAB}})}, \tag{8}$$

where $\mathcal{S}$ is always limited to sites without polymorphism in the outgroup O. This statistic is in principle capturing the same source of asymmetry as the statistic proposed by *Hahn and Hibbins, 2019*, although their version uses divergence to the outgroup instead of frequencies of site-counts. Similarly, the four-species $D_4$ statistic, which considers species tree {{{P$_1$,P$_2$},O$_1$},O$_2$} and site patterns ABBA versus BABA (*Durand et al., 2011*) is calculated as

$$D_{\text{P}_1,\text{P}_2,\text{O}_1,\text{O}_2} = \frac{\sum_{s\in\mathcal{S}}(f_{s,\text{ABBA}} - f_{s,\text{BABA}})}{\sum_{s\in\mathcal{S}}(f_{s,\text{ABBA}} + f_{s,\text{BABA}})}, \tag{9}$$

where $\mathcal{S}$ is always limited to sites without polymorphism in the second outgroup O$_2$. The significance of both tests was computed using block-jackknife over 1 Mb blocks across the genome. Additionally, we estimated rate ratio as follows. First we restricted to sites where all outgroups are fixed for the same ancestral allele to dampen the influence of recurrent mutation. Then, for each site, sample one allele at random from each focal lineage. Calculate the probability of observing a derived allele in P$_1$ but not in P$_2$, and the probability of observing a derived allele in P$_2$ but not in P$_1$. The rate ratio is computed as the ratio between the two probabilities. Explicitly, let $I(\cdot)$ be the identity function, and $f_s$ be the frequency of the derived allele, then:

$$\text{Rate ratio } r = \frac{\sum_{s\in\mathcal{S}} f_{s,\text{P}_1}(1 - f_{s,\text{P}_2})\Pi_{i\in\text{outgroups}}I(f_{s,i} = 0)}{\sum_{s\in\mathcal{S}}(1 - f_{s,\text{P}_1})f_{s,\text{P}_2}\Pi_{i\in\text{outgroups}}I(f_{s,i} = 0)} \tag{10}$$

Its standard error was estimated using 1 Mb block-jackknifing. We excluded *P. xuthus* from the outgroups to increase the number of informative sites when using this formula.

## $D_3$ and $D_4$ under unequal substitution rates and recurrent mutations

In this section, we calculate observed $D_3$ and $D_4$ assuming that incomplete lineage sorting contributes insignificantly to both statistics. If incomplete lineage sorting is present, it will not create new bias (numerators are on average unchanged), but will likely dampen existing bias (inflating denominators).

As substitutions are independent along each lineage, we can mute recurrent mutations in outgroups and generate them afterwards. For three taxa with gene tree {{P$_1$,P$_2$},O$_1$}, before recurrent mutations, there are $n_1$ sites with pattern (B,A,A), and $n_2$ sites with pattern (A,B,A). If substitution rate is higher in P$_2$, we have $n_2 > n_1$, so the true value of $D_3$, written as $\widehat{D}_3$, is always negative:

$$\widehat{D}_3 = \frac{n_1 - n_2}{n_1 + n_2} < 0 \tag{11}$$

Next, recurrent mutations in O$_1$ occur at each site with an average probability $p_1$, and with an average probability $c$, ancestral alleles from affected sites in O$_1$ are converted to the same derived allele in {P$_1$,P$_2$}. $c$ will be independent of $n_1$ and $n_2$, as long as substitutions between {P$_1$,P$_2$} are only different in *rates*, but not mutation types. Hence, two possible mutation paths exist:

$$(\text{A}, \text{B}, \text{B}) \xrightarrow{p_1} (\text{A}, \text{B}, \cdot) \xrightarrow{c} (\text{A}, \text{B}, \text{A}) \equiv (\text{B}, \text{A}, \text{B})$$
$$(\text{B}, \text{A}, \text{B}) \xrightarrow{p_1} (\text{B}, \text{A}, \cdot) \xrightarrow{c} (\text{B}, \text{A}, \text{A}) \equiv (\text{A}, \text{B}, \text{B}) \tag{12}$$

The expected site counts, after recurrent mutations, become

$$\langle n_{\text{ABB}} \rangle = (1 - p_1)n_1 + cp_1 n_2$$
$$\langle n_{\text{BAB}} \rangle = (1 - p_1)n_2 + cp_1 n_1 \tag{13}$$

Using the new expected site counts in $D_3$ statistics produce the following value:

$$D_3 = \frac{1 - (c + 1)p_1}{1 + (c - 1)p_1} \widehat{D}_3 \approx (1 - 2cp_1)\widehat{D}_3 \tag{14}$$

Since $\widehat{D}_3$ is negative, it grows approximately linearly near small values of $p_1$. (The full equation is still monotonic in $p_1$.)

Similarly, for four-taxon statistics, before recurrent mutation, there are two types of sites: (A,B,B,B)- $n_1$; (B,A,B,B)- $n_2$. Suppose the average probability of recurrent mutation is $p_1$ in $O_1$, and $p_2$ in $O_2$, and the conversion probability of each recurrent mutation into derived alleles of $\{P_1, P_2\}$ is $c$ for both outgroups. Using the same procedure, one can show that

$$D_4 = \frac{p_2 - p_1}{p_2 + p_1} \widehat{D}_3 \tag{15}$$

Since recurrent mutations occur more frequently in distant outgroups, $p_2 > p_1$. Because $\widehat{D}_3$ is negative, we have $D_4 < 0$.

## Local gene trees

Local gene trees were estimated using iqtree-2.0 (**Minh et al., 2020**) on 50 kb non-overlapping genomic windows with options -m MFP -B 5000. Only SNPs from annotated regions (synonymous sites +nonsynonymous sites +introns) across all individuals were used. For diploid individuals, heterozygous sites were assigned IUPAC ambiguity codes and iqtree assigned equal likelihood for each underlying character, thus information from heterozygous sites is largely retained. This is crucial as we are interested in the branch length of inferred trees. Option -m MFP implements iqtree's ModelFinder that tests the FreeRate model to accommodate maximum flexibility with rate-variation among sites (**Kalyaanamoorthy et al., 2017**). We also used UltraFast Bootstrap to calculate the support for different types of splits in each window (the -B 5000 option; **Hoang et al., 2018**). In each window, we extracted the support for monophyly among *P. maackii* +*P.syfanius* directly from the output of UltraFast Bootstrap, and we define the support for paraphyly among *P. maackii* +*P.syfanius* as (100 - the support for monophyly). For each level of support, we filtered out genomic windows where both the support for monophyly and the support for paraphyly drop below the given level. The remaining windows were considered informative.

## Rate-mixing under the equilibrium IM model

We construct a continuous-time coalescent model as follows. Both populations have $N$ haploid individuals, gene flow rate is $m$, and coalescent rate is $N^{-1}$ in each population. In the equilibrium system, as we track both haploid individuals backward in time, there are six distinct states: $(1|2)$, $(2|1)$, $(1, 2|)$, $(|1, 2)$, $(0|)$, $(|0)$, where 1 and 2 represent two individuals prior to coalescent, 0 is the state of coalescent, and $(\cdot|\cdot)$ shows the location of each lineage. Its transition density $\mathbf{p}(t)$ satisfies $\partial_t \mathbf{p} = \mathbf{A}\mathbf{p}$, where $\mathbf{A}$ is given as

$$\begin{pmatrix} -2m & 0 & m & m & 0 & 0 \\ 0 & -2m & m & m & 0 & 0 \\ m & m & -2m - \frac{1}{N} & 0 & 0 & 0 \\ m & m & 0 & -2m - \frac{1}{N} & 0 & 0 \\ 0 & 0 & \frac{1}{N} & 0 & -m & m \\ 0 & 0 & 0 & \frac{1}{N} & m & -m \end{pmatrix} \tag{16}$$

Let $S_{i|j}(T)$ be the mean sojourn time of an uncoalesced individual inside population during $0 \le t \le T$, conditioning on the individual being taken from population $j$ at $t = 0$. Assuming the

infinite-site mutation model, let $\mu_i$ be the substitution rate in population , observed rate ratio $r$ is thus

$$r = \frac{\mu_2 S_{2|2}(\infty) + \mu_1 S_{1|2}(\infty)}{\mu_1 S_{1|1}(\infty) + \mu_2 S_{2|1}(\infty)} \tag{17}$$

where (due to symmetry)

$$S_{1|1}(\infty) = S_{2|2}(\infty) = \int_0^{+\infty} (1,0,0,1,0,0) e^{At}(1,0,0,0,0,0)^\mathsf{T} dt = \frac{1+2Nm}{2m}$$
$$S_{2|1}(\infty) = S_{1|2}(\infty) = \int_0^{+\infty} (0,1,1,0,0,0) e^{At}(1,0,0,0,0,0)^\mathsf{T} dt = N \tag{18}$$

Let $r_0 = \mu_2/\mu_1$, and since $F_{ST} = (1 + 4Nm)^{-1}$, we have

$$r = \frac{1 + r_0 + F_{ST}(r_0 - 1)}{1 + r_0 - F_{ST}(r_0 - 1)} \tag{19}$$

## Local ancestry estimation

Software ELAI (*Guan, 2014*) with a double-layer HMM model was used to estimate diploid local ancestries across chromosomes. An example command is as follows: elai-lin -g genotype.maackii.txt -p 10 -g genotype.syfanius.txt -p 11 -g genotype.admixed.txt -p 1 -pos position.txt -s 30 -C 2 -c 10 -mg 5000 --exclude-nopos.

Note that -mg specifies the resolution of ancestry blocks, thus increasing its value will increase the stochastic error of incorrectly inferring very short blocks of ancestry. To control for uncertainty, we estimated repeatedly for 50 times. All replicates were used simultaneously in finding the correlation coefficients between entropy and other variables. Results from an example run is in *Figure 3—figure supplement 1*.

## Ancestry and entropy

Here we introduce concisely the data transformation framework for calculating the entropy of local ancestry. The mathematical detail of this approach is presented in the appendix.

### Ancestry representation

The space of all ancestry signals is high-dimensional, and directly calculating the entropy in this space is not feasible with just a few individuals. So we propose to measure only the pairwise correlation of ancestries among sites, which captures only the second-order randomness, but is sufficient for practical purposes. Consider a hybrid individual with two parental populations indexed by $k = 1, 2$. Assuming a continuous genome, let $p_k(l) = 0, \frac{1}{2}, 1$ be the diploid ancestry of locus $l$ within genomic interval $[0, L]$. By definition, we have $p_1(l) + p_2(l) = 1$, that is the total ancestry is conserved everywhere in the genome. The bi-ancestry signal at locus $l$ is defined as the following complex variable

$$z(l) = \sqrt{p_1(l)} + i\sqrt{p_2(l)} = e^{i \arccos \sqrt{p_1(l)}}, \tag{20}$$

where $i = \sqrt{-1}$ is the imaginary number. An advantage of using a complex representation for the bi-ancestry signal is that we can model different ancestries along the genome as different phases of a complex unit phasor ($e^{i\theta}$), such that the power of the signal at any given locus is simply the sum of both ancestries, which is conserved ($|z(l)|^2 = 1$). It ensures that we do not bias the analysis to any particular region or any particular individual when decomposing the signal into its spectral components.

### Within-individual spectral entropy ($S_w$)

To characterize the average autocorrelation along an ancestry signal at a given scale $l$, define the following scale-dependent autocorrelation function

$$A(l) = \mathrm{Re}[\tfrac{1}{L} \int_0^L z(\xi)\overline{z(\xi + l)}\, d\xi], \tag{21}$$

where $z(l)$ is understood as a periodic function such that $z(\xi + l) = z(\xi + l - L)$ whenever the position goes outside of $[0, L]$. The Wiener-Khinchin theorem guarantees that $z(l)$'s power spectrum $\zeta(f)$, which is discrete, and the autocorrelation function $A(l)$ form a Fourier-transform pair. Due to the uncertainty principle of Fourier transform, $A(l)$ that vanishes quickly at short distances (small-scale autocorrelation) will produce a wide $\zeta(f)$, and vice versa. So the entropy $S_w$ of $\zeta(f)$, which measures the spread of the total ancestry into each spectral component, also measures the scale of autocorrelation. In practice, $\zeta(f)$ is the square modulus of the Fourier series coefficients of $z(l)$, and we fold the spectrum around $f = 0$ before calculating the within-individual entropy $S_w$. The formula used in the manuscript is

$$S_w = -\sum_{n=0}^{+\infty} \zeta_n \ln \zeta_n$$
$$\zeta_n = \begin{cases} |Z_n|^2 + |Z_{-n}|^2 & (n > 0) \\ |Z_0|^2 & (n = 0) \end{cases} \tag{22}$$

where $Z_n$ are the Fourier coefficients from the expansion $z(l) = \sum_{n=-\infty}^{+\infty} Z_n e^{i2\pi nl/L}$. To speed up the Fourier expansion, we could *densely* pack equally-spaced markers that sample a continuous ancestry signal into a discrete signal, which then undergoes Fast Fourier Transform (FFT). The spectrum of FFT (discrete and finite) approximates the continuous-time Fourier spectrum (discrete and infinite), and entropy also converges as marker density increases.

## Between-individual spectral entropy ($S_b$)

As ancestry configuration is far from random around barrier loci, it will also influence the correlation of ancestry between different individuals at the same locus. For a genomic region experiencing strong barrier effects, two individuals could either be very similar in ancestry, or very different. This effect can be quantified by first calculating the cross-correlation $C_{j,j'}(l) = z_j(l)\overline{z_{j'}(l)}$ at position $l$ between individuals $j$ and $j'$, and then averaging across a genome interval: $c_{j,j'} = \frac{1}{L} \int_0^L C_{j,j'}(l)\,dl$. The $J \times J$ dimensional matrix $\mathbf{C}$ with entries $c_{j,j'}$ describes the pairwise cross-correlation within the cohort of $J$ individuals. We also have $c_{j,j} \equiv 1$ as each individual is perfectly correlated with itself. The matrix $\mathbf{C}$ is Hermitian, so it has a real spectral decomposition with eigenvalues $\lambda_j$ that satisfy $\sum_j \lambda_j/J = 1$. This process is very similar to performing a principal component analysis on the entire cohort of individuals, and $\lambda_j/J$ describes the fraction of the total ancestry projected onto principal component $j$. If many loci co-vary in ancestry, the spectrum $\{\lambda_j\}$ will be concentrated near the first few components. Similarly, we use entropy to measure the spread of the spectrum, and hence the between-individual spectral entropy is defined as

$$S_b = -\sum_j \frac{\lambda_j}{J} \ln \frac{\lambda_j}{J} \tag{23}$$

## Acknowledgements

This work was supported by the Lewis and Clark Fund for Exploration and Field Research (American Philosophical Society, 2017–2018) granted to TX.; TX is also funded by a studentship from Harvard Department of Organismic and Evolutionary Biology, the NSF-Simons Center for Mathematical and Statistical Analysis of Biology at Harvard (award number #1764269) and the Harvard Quantitative Biology Initiative during the project. We thank Harvard FAS Research Computing for providing computation resources. We thank Fernando Seixas for discussion on hybrid ancestry inference. We thank Kaifeng Bu for his comments on information-theoretic measures. We thank Nathaniel Edelman, Neil Rosser, Miriam Miyagi, Yuttapong Thawornwattana, Sarah Dendy, Liang Qiao, Adam Cotton, John Wakeley, Robin Hopkins, and Naomi Pierce for their valuable input. We also thank Yuchen Zheng, Zhuoheng Jiang, Shaoji Hu, Feng Cao, and Shui Xu for support in the field. Feng Cao provided specimen photos from the hybrid zone.

## Additional information

### Funding

| Funder | Grant reference number | Author |
|---|---|---|
| American Philosophical Society | The Lewis and Clark Fund for Exploration and Field Research (2017-2018) | Tianzhu Xiong |
| The NSF-Simons Center for Mathematical and Statistical Analysis of Biology (Award Number #1764269), and The Quantitative Biology Initiative (Harvard University) | Graduate Student Fellowship | Tianzhu Xiong |
| Department of Organismic and Evolutionary Biology, Harvard University | Graduate Student Fellowship | Tianzhu Xiong |
| Department of Organismic and Evolutionary Biology, Harvard University | Faculty Start-up Fund | James Mallet |

The funders had no role in study design, data collection and interpretation, or the decision to submit the work for publication.

### Author contributions

Tianzhu Xiong, Conceptualization, Data curation, Formal analysis, Funding acquisition, Investigation, Methodology, Project administration, Resources, Software, Visualization, Writing - original draft, Writing – review and editing; Xueyan Li, Resources, The reference genomes for P. bianor and P. xuthus. Fieldworks, Writing – review and editing; Masaya Yago, Museum specimens of P. syfanius and P. maackii, Resources, Writing – review and editing; James Mallet, Conceptualization, Funding acquisition, Project administration, Resources, Supervision, Writing - original draft, Writing – review and editing

### Author ORCIDs

Tianzhu Xiong ⬦ http://orcid.org/0000-0002-4576-8764
James Mallet ⬦ http://orcid.org/0000-0002-3370-0367

### Decision letter and Author response

Decision letter https://doi.org/10.7554/eLife.78135.sa1
Author response https://doi.org/10.7554/eLife.78135.sa2

---

# Additional files

### Supplementary files

• Transparent reporting form

### Data availability

Source code is available at: https://github.com/tzxiong/2021_Maackii_Syfanius_HybridZone, (copy archived at swh:1:rev:41b220f489e17ff9795e5a0666e9579a00b2b3b8) Whole-genome sequences are deposited in the National Center for Biotechnology Information, Sequence Read Archive (BioProject Accession Number: PRJNA765117).

The following datasets were generated:

| Author(s) | Year | Dataset title | Dataset URL | Database and Identifier |
|---|---|---|---|---|
| Xiong T, Mallet J | 2022 | Hybridization between Papilio syfanius and Papilio maackii | https://www.ncbi.nlm.nih.gov/sra/PRJNA765117 | NCBI Sequence Read Archive, PRJNA765117 |
| Xiong T | 2022 | Database for the code used in the 2021 manuscript on the hybrid zone between Papilio syfanius and Papilio maackii | https://github.com/tzxiong/2021_Maackii_Syfanius_HybridZone | GitHub, 2021_Maackii_Syfanius_HybridZone |

The following previously published dataset was used:

| Author(s) | Year | Dataset title | Dataset URL | Database and Identifier |
|---|---|---|---|---|
| Lu S, Yang J, Dai X, Xie F, He J, Dong Z, Mao J, Liu G, Chang Z, Zhao R, Wan W, Zhang R, Li Y, Wang W LX | 2019 | Supporting data for "Chromosomal-level reference genome of Chinese peacock butterfly (Papilio bianor) based on third-generation DNA sequencing and Hi-C analysis" | http://dx.doi.org/10.5524/100653 | GigaDB Datasets, 10.5524/100653 |

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

## Appendix 1

### Representation of ancestry on a hybrid chromosome

The ancestry of a hybrid depends on the pure reference populations. The following assumptions are used throughout this section:

- There are a finite number of pure reference populations indexed by $k \in \{1, 2, \cdots, K\}$. In most cases, we are only interested in $K = 2$, for instance, a hybrid zone between two lineages.
- The chromosome has so many sites so that it can be treated as a contiguous rod with length $L$. $l \in [0, L]$ is the index of positions on a chromosome.
- The species' ploidy is $n_p$ (completely phased data $\Leftrightarrow n_p = 1$). When not specified, we assume the data is unphased, so that the ancestry on a particular chromosome always refers to the ancestry on a collection of $n_p$ homologous chromosomes.
- Ancestry can be inferred. In practice, regions with low density of informative SNPs will make inference difficult.

As "ancestry" is a categorical variable (i.e., there is no intrinsic order among the reference populations), to quantify the correlation of ancestry along a chromosome, we need to map the contribution of each ancestry category to some numeric value that can be used to calculate correlation.

The first choice is to use the probability of each category. For unphased data, let $p_k(l)$ be the fraction of the $n_p$-ploid chromosome at position $l$ coming from reference population $k$. The vector-valued function $\mathbf{p}(l) = (p_1(l), p_2(l), \cdots, p_K(l))^\top$ completely describes the ancestry on a single chromosome. By definition, $\sum_k p_k(l) \equiv 1$. This is the conservation of total ancestry at any position. However, by this representation, the correlation $\mathbf{p}^\top(l_1)\mathbf{p}(l_2)$, which is the product of ancestries at different positions, will have units of the square of probabilities. If a locus is to be compared with itself, then the correlation will be

$$\mathbf{p}^\top(l)\mathbf{p}(l) = \sum_k p_k^2(l) \leq 1 \tag{24}$$

Ideally, we want the correlation of ancestry of a locus with itself to be the same regardless of its ancestry configuration. *Equation (24)* does not meet this criteria as it depends on $\mathbf{p}$. Here we borrow some concepts from signal-processing theory. The conservation of total ancestry means that ancestry is more analogous to the "energy" of a signal, which is in units of [squared-signal], rather than the signal itself. Therefore, we could instead define a second type of ancestry representation by taking the square roots of probabilities.

### Definition 1 (Spherical representation of ancestry)

If the probability representation of ancestry is $\mathbf{p}(l)$, then the spherical representation $\mathbf{y}(l)$ takes the square-root of each element of $\mathbf{p}(l)$:

$$\mathbf{y}(l) = (\sqrt{p_1(l)}, \sqrt{p_2(l)}, \cdots, \sqrt{p_K(l)})^\top \tag{25}$$

Following this representation, the autocorrelation between two spherical ancestries becomes a natural measure of the similarity between two probability distributions:

### Definition 2 (Correlation between two spherical ancestries)

The autocorrelation function between $\mathbf{y}(l_1)$ and $\mathbf{y}(l_2)$ is the following quantity known as the Bhattacharyya coefficient, also known as the fidelity measure in information theory:

$$A(l_1, l_2) = \mathbf{y}^\top(l_1)\mathbf{y}(l_2) = \sum_k \sqrt{y_k(l_1)y_k(l_2)} \in [0, 1] \tag{26}$$

Further, self-correlation is always 1:

$$A(l, l) = \mathbf{y}^\top(l)\mathbf{y}(l) = \sum_k p_k(l) \equiv 1 \tag{27}$$

The conservation of self-correlation will be important when we decompose ancestry into its spectral components, as it will not bias our analysis to any particular region of the chromosome.

There is also a geometric meaning associated with this representation. Since $\mathbf{y}(l)$ has a $L^2$-norm of 1, each ancestry configuration corresponds to a point on the unit sphere in $\mathbb{R}^K$, and different ancestries are represented by the orientation of this vector in $\mathbb{R}^K$.

In many studies we are dealing only with two reference populations, such as hybrid zones between a pair of divergent lineages. This situation allows a more compact representation of ancestry using complex numbers:

### Definition 3 (Complex representation of ancestry)
If $K = 2$, define the following complex variable $z(l)$ as the representation of ancestries:

$$z(l) = \sqrt{p_1(l)} + i\sqrt{p_2(l)} \tag{28}$$

where $i = \sqrt{-1}$ is the imaginary number.

This seemingly artificial definition is not the first time that a complex number is used to model a physical phenomenon. In quantumn mechanics, the quantumn wave function of a particle is represented by a complex wave with the probability of occurrence measured by the square-modulus of the wave. Here, we are also using the square-modulus of $z(l)$ to represent the total ancestry of a given locus.

A complex-valued signal $z(l)$, like any real signal, has the definition of autocorrelation:

### Definition 4 (Correlation between two complex ancestries)
The correlation between two complex ancestries $z(l_1)$ and $z(l_2)$ is the following product:

$$
\begin{aligned}
A(l_1, l_2) &= z(l_1)\overline{z(l_2)} \\
&= \sqrt{p_1(l_1)p_1(l_2)} + \sqrt{p_2(l_1)p_2(l_2)} + i(\sqrt{p_2(l_1)p_1(l_2)} - \sqrt{p_1(l_1)p_2(l_2)})
\end{aligned}
\tag{29}
$$

By this definition, the real part $\mathrm{Re}[A(l_1, l_2)]$ of the autocorrelation is just the Bhattacharyya coefficient between two ancestry configurations on $l_1$ and $l_2$, which is a measure of similarity. The absolute value of the imaginary part of the autocorrelation $|\mathrm{Im}[A(l_1, l_2)]|$ measures the volume of the parallelogram spanned by vectors $\mathbf{y}(l_1)$ and $\mathbf{y}(l_2)$, thus it is a measure of dissimilarity. Further, the self-correlation $A(l, l)$ is also conserved for all $l$, as $\mathrm{Im}[A(l, l)] \equiv 0$ and $\mathrm{Re}[A(l, l)] \equiv 1$.

The complex representation possesses some useful properties not shared with the spherical representation, but it also limits our analysis to cases with only two reference populations. This will not pose a problem for the study of hybrid zones between a pair of parapatric lineages.

## Mean autocorrelation vs. hybrid index
The hybrid index is defined as the average ancestry in a hybrid from a given reference population over a set of loci. It is usually calculated for each individual or for each chromosome. In our notation:

### Definition 5 (Hybrid index)
The hybrid index over a given genomic interval $[0, L]$ for reference population $k$ is

$$h_k = \frac{1}{L} \int_0^L p_k(l) \, \mathrm{d}l \tag{30}$$

In contrast, the average of the spherical or the complex representation provides information about autocorrelation within the chromosome instead of the mean ancestry.

### Theorem 1 (Mean autocorrelation)
With the spherical or the complex representation, the squared $L^2$-norm (or the squared modulus, when a complex representation is available) of the average signal represents the mean autocorrelation of the ancestry on the genomic interval $[0, L]$.

$$a := \frac{1}{L^2} \iint_{[0,L]^2} A(l_1, l_2) \, \mathrm{d}l_1 \, \mathrm{d}l_2 = \left\| \frac{1}{L} \int_0^L \mathbf{y}(l) \, \mathrm{d}l \right\|^2 = \left| \frac{1}{L} \int_0^L z(l) \, \mathrm{d}l \right|^2 \tag{31}$$

Proof. For a general spherical representation $\mathbf{y}(l)$, we have

$$\left\| \frac{1}{L} \int_0^L \mathbf{y}(l) \mathrm{d}l \right\|^2 = \frac{1}{L^2} \sum_k \left( \int_0^L \sqrt{p_k(l)} \mathrm{d}l \right) = \frac{1}{L^2} \sum_k \int_0^L \sqrt{p_k(l_1)} \mathrm{d}l_1 \int_0^L \sqrt{p_k(l_2)} \mathrm{d}l_2$$
$$= \frac{1}{L^2} \iint_{[0,1]^2} \sum_k \sqrt{p_k(l_1) p_k(l_2)} \, \mathrm{d}l_1 \, \mathrm{d}l_2 = \frac{1}{L^2} \iint_{[0,L]^2} A(l_1, l_2) \, \mathrm{d}l_1 \, \mathrm{d}l_2$$

For a complex representation $z(l)$, notice that $A(l_1, l_2) = \overline{A(l_2, l_1)}$, so

$$\frac{1}{L^2} \iint_{[0,L]^2} A(l_1, l_2) \, \mathrm{d}l_1 \, \mathrm{d}l_2 = \frac{1}{L^2} \iint_{[0,L]^2} \mathrm{Re} A(l_1, l_2) \, \mathrm{d}l_1 \, \mathrm{d}l_2 = \frac{1}{L^2} \iint_{[0,1]^2} \sum_{k=1,2} \sqrt{p_k(l_1) p_k(l_2)} \, \mathrm{d}l_1 \, \mathrm{d}l_2$$

This guarantees that the mean autocorrelation when using a complex representation is the same as the mean autocorrelation when using a spherical representation with $K = 2$. Additionally,

$$\left| \frac{1}{L} \int_0^L z(l) \mathrm{d}l \right|^2 = \frac{1}{L^2} \int_0^L z(l_1) \mathrm{d}l_1 \overline{\int_0^L z(l_2) \mathrm{d}l_2} = \frac{1}{L^2} \iint_{[0,L]^2} z(l_1) \overline{z(l_2)} \mathrm{d}l_1 \mathrm{d}l_2 = \frac{1}{L^2} \iint_{[0,L]^2} A(l_1, l_2) \mathrm{d}l_1 \mathrm{d}l_2$$

This completes the proof.

The quantity $a$ is a measure of the average similarity of the ancestry configurations. It does not consider the genomic position of different ancestry configurations, so it does not contain information about whether similar ancestry configurations are clustered together.

## The entropy of ancestry

To characterize the scale of correlation, the distance between loci is important because correlation often drops while distance between loci increases. The information about the spatial scale of correlation is retained when the full spectrum of correlation is considered within a single individual.

A second source of correlation arises when we consider the relationship between individuals. To convey the main idea, let's compare between a set of ancestry signals from an inversion and a set of ancestry signals from a regular region subject to normal recombination. For the inversion, since recombination between chromosomes of different ancestry is often completely suppressed, the ancestry signal along the inversion will be close to constant. When two haploid individuals are compared at the inverted region, they are either different everywhere in terms of ancestry, or completely the same. Comparing between diploid individuals is similar, although the difference is more fine-grained due to the presence of the heterozygotes. However, in a collinear region subject to a regular rate of recombination, any ancestry signal will switch stochastically between states. In the latter case, the similarity between two ancestry signals will be similar across multiple pairwise comparisons. It will be helpful to think in the principal component (PC) space spanned by all individuals' ancestry signals. Ancestry signals from an inversion will form several tight clusters in the PC space, while those from a regular region will form a single cloud with a larger dispersion.

Both sources of correlation, and hence both types of randomness, can be measured using entropy in information theory.

### Definition 6 (Shannon entropy)

The Shannon entropy for a discrete probability distribution $\{p_j\}$, $j \in \mathbb{Z}$, is defined as

$$S(\{p_j\}) = - \sum_{j \in \mathbb{Z}} p_j \ln p_j \tag{32}$$

For a continuous distribution with probability density function $p(x)$, $x \in \mathbb{R}$, the Shannon differential entropy is defined as

$$S(p(x)) = - \int_{\mathbb{R}} p(x) \ln p(x) \, \mathrm{d}x \tag{33}$$

For any nonnegative series $\{p_j\}$ or nonnegative function $p(x)$, suppose they converge upon summation/integration, we use the same notations $S(\{p_j\})$ and $S(p(x))$ to denote the entropy after they are appropriately normalized to probability distributions.

Shannon entropy (or any other entropy measure) is a useful measure of the spread of a distribution over its entire configuration space. When the distribution is concentrated (low randomness, high certainty), $S$ will be low. $S = 0$ if and only if $p_j = 1$ for some $j$.

## Entropy associated with the correlation within a single individual

### Definition 7 (Fourier spectrum of the spherical representation)

For the spherical representation of ancestry on genomic interval $[0, L]$

$$\mathbf{y}(l) = (\sqrt{p_1(l)}, \sqrt{p_2(l)}, \cdots, \sqrt{p_K(l)})^\top,$$

expand each component into its Fourier series:

$$\sqrt{p_k(l)} = \sum_{n=-\infty}^{+\infty} \hat{p}_{k,n} e^{i\frac{2\pi}{L}nl}$$

The folded Fourier spectrum is defined as

$$\zeta_n = \begin{cases} 2\sum_{k=1}^{K} |\hat{p}_{k,n}|^2 & (n > 0) \\ \sum_{k=1}^{K} |\hat{p}_{k,0}|^2 & (n = 0) \end{cases} \tag{34}$$

### Definition 8 (Fourier spectrum of the complex representation)

For the complex representation of ancestry on genomic interval $[0, L]$

$$z(l) = \sqrt{p_1(l)} + i\sqrt{p_2(l)},$$

expand $z(l)$ into its Fourier series:

$$z(l) = \sum_{n=-\infty}^{+\infty} Z_n e^{i\frac{2\pi}{L}nl}$$

The folded Fourier spectrum is defined as

$$\zeta_n = \begin{cases} |Z_n|^2 + |Z_{-n}|^2 & (n > 0) \\ |Z_0|^2 & (n = 0) \end{cases} \tag{35}$$

The following theorem guarantees that the folded spectrum $\zeta_n$ is the same for bi-ancestry signals using either representation.

### Theorem 2

When $K = 2$, the folded Fourier spectrum $\zeta_n$ is the same for $\mathbf{y}(l) = (\sqrt{p_1(l)}, \sqrt{p_2(l)})^\top$ and $z(l) = \sqrt{p_1(l)} + i\sqrt{p_2(l)}$.

Proof. Expand each component into its Fourier series:

$$\sqrt{p_1(l)} = \sum_{n=-\infty}^{+\infty} \hat{p}_{1,n} e^{i\frac{2\pi}{L}nl}$$
$$\sqrt{p_2(l)} = \sum_{n=-\infty}^{+\infty} \hat{p}_{2,n} e^{i\frac{2\pi}{L}nl}$$

The folded spectrum using the spherical representation is

$$\zeta_n = \begin{cases} 2|\hat{p}_{1,n}|^2 + 2|\hat{p}_{2,n}|^2 & (n > 0) \\ |\hat{p}_{1,0}|^2 + |\hat{p}_{2,0}|^2 & (n = 0) \end{cases}$$

Using the linearity of Fourier expansion, we have

$$Z_n = \hat{p}_{1,n} + i\hat{p}_{2,n}$$

Thus,

$$
\begin{aligned}
Z_n \overline{Z_n} &= (\hat{p}_{1,n} + i\hat{p}_{2,n})(\overline{\hat{p}_{1,n}} - i\overline{\hat{p}_{2,n}}) = \hat{p}_{1,n}\overline{\hat{p}_{1,n}} + \hat{p}_{2,n}\overline{\hat{p}_{2,n}} + i\hat{p}_{2,n}\overline{\hat{p}_{1,n}} - i\hat{p}_{1,n}\overline{\hat{p}_{2,n}} \\
&= \hat{p}_{1,n}\overline{\hat{p}_{1,n}} + \hat{p}_{2,n}\overline{\hat{p}_{2,n}} - 2\mathrm{Im}(\hat{p}_{2,n}\overline{\hat{p}_{1,n}})
\end{aligned}
$$

Since the Fourier coefficients of a real function at opposite frequencies are complex conjugates to each other, we have

$$
Z_{-n}\overline{Z_{-n}} = \hat{p}_{1,-n}\overline{\hat{p}_{1,-n}} + \hat{p}_{2,-n}\overline{\hat{p}_{2,-n}} - 2\mathrm{Im}(\hat{p}_{2,-n}\overline{\hat{p}_{1,-n}}) = \overline{\hat{p}_{1,n}}\hat{p}_{1,n} + \overline{\hat{p}_{2,n}}\hat{p}_{2,n} - 2\mathrm{Im}(\overline{\hat{p}_{2,n}}\hat{p}_{1,n})
$$

Finally,

$$
|Z_n|^2 + |Z_{-n}|^2 = Z_n\overline{Z_n} + Z_{-n}\overline{Z_{-n}} = 2\hat{p}_{1,n}\overline{\hat{p}_{1,n}} + 2\hat{p}_{2,n}\overline{\hat{p}_{2,n}} - 2\mathrm{Im}(\hat{p}_{2,n}\overline{\hat{p}_{1,n}} + \overline{\hat{p}_{2,n}}\hat{p}_{1,n})
$$

Since $\hat{p}_{2,n}\overline{\hat{p}_{1,n}} + \overline{\hat{p}_{2,n}}\hat{p}_{1,n}$ is real, the last term of the previous equation becomes zero. For the zero-th component $\zeta_0$, as both $\hat{p}_{1,0}$ and $\hat{p}_{2,0}$ are real, there is no difference between the two representations. This completes the proof.

A nice property of the Fourier spectrum $\zeta_n$ is that it can be interpreted as a probability distribution of ancestry among different frequency components. By Parseval's theorem, it is easy to verify that $\sum_n \zeta_n = \frac{1}{L}\int_0^L (\sum_k p_k(l))\,dl = 1$. From the Wiener-Khinchin theorem, the unfolded spectrum forms a Fourier transform pair with the autocorrelation function of the original signal. The significance of the Wiener-Khinchin theorem is that information about the autocorrelation of the original signal can now be extracted from the Fourier spectrum $\zeta_n$.

## Definition 9 (Within-individual entropy)

Let $S_w = -\sum_n \zeta_n \ln \zeta_n$. $S_w$ is the Shannon entropy of the folded Fourier spectrum $\zeta_n$. As $S_w$ captures the correlation structure within each individual, we also call it the within-individual entropy.

Formally, we have the following uncertainty principle relating the within-individual entropy to the scale of autocorrelation.

## Theorem 3 (Entropic uncertainty)

Let $A(l) = \frac{1}{L}\int_0^L \mathbf{y}^\top(x)\mathbf{y}(x+l)\,dx$ be the average autocorrelation at scale $l$ ($0 \le l \le L$) for the spherical ancestry. Here, $\mathbf{y}$ is understood as a periodic function of period $L$. The following inequality holds:

$$
S_w = S(\{\zeta_n\}) \ge \int_0^L \frac{A^2(l)}{Q}\ln\frac{A^2(l)}{Q}\,dl + \left(\frac{1}{L}\int_0^L A(l)\,dl - 1\right)\ln 2 + \ln L \tag{36}
$$

where $Q = \int_0^L A^2(l)\,dl$ is a normalization factor. Note that the right-hand-side of the inequality is invariant under linearly re-scaling $A(l)$ to a different interval $[0, L']$. Thus, we can also write the inequality compactly, supposing $A(l)$ has been rescaled to $[0, 2]$, as

$$
S_w \ge -S(A^2(l)) + a\ln 2, \tag{37}
$$

where $a$ is the average autocorrelation defined in *Equation 31*.

Proof. (i) In this part, we establish the Wiener-Khinchin relation that $A(l)$ expands into a Fourier series with coefficients $\eta_n = \sum_{k=1}^K |\hat{p}_{k,n}|^2$:

$$
A(l) = \sum_{n=-\infty}^{+\infty} \sum_{k=1}^K |\hat{p}_{k,n}|^2 e^{i\frac{2\pi}{L}nl} \tag{38}
$$

The derivation is as follows:

$$
\begin{aligned}
A(l) &= \frac{1}{L}\int_0^L \sum_{K=1}^K y_k(x)y_k(x+1)\,dx = \sum_{K=1}^K \frac{1}{L}\int_0^L \left(\sum_{n=-\infty}^{+\infty} \hat{P}_{k,n}e^{i\frac{2\pi}{L}nx}\right)\left(\sum_{n=-\infty}^{+\infty} \hat{P}_{k,n}e^{i\frac{2\pi}{L}n(x+1)}\right)dx \\
&= \sum_{K=1}^K \sum_{n=-\infty}^{+\infty} e^{i\frac{2\pi}{L}nl}\frac{1}{L}\int_0^L \hat{P}_{k,n}\hat{P}_{k,-n}\,dx = \sum_{n=-\infty}^{+\infty}\sum_{K=1}^K \left|\hat{P}_{k,n}\right|^2 e^{i\frac{2\pi}{L}nl} = \sum_{n=-\infty}^{+\infty} \eta_n e^{i\frac{2\pi}{L}nl}
\end{aligned} \tag{39}
$$

(ii) Let $\int_0^L A^2(l)\,dl = Q$. It is obvious that $\{\eta_n/\sqrt{Q}\}$ are Fourier series coefficients of $A(l)/\sqrt{Q}$, they obey the Hausdorff-Young inequality

$$\left( \sum_{n=-\infty}^{+\infty} \left|\eta_n/\sqrt{Q}\right|^{q'} \right)^{\frac{1}{q'}} \leq \left( \int_0^1 \left|A(xL)/\sqrt{Q/L}\right|^q dx \right)^{\frac{1}{q}}, \tag{40}$$

where $q \in (1,2)$ and $1/q' + 1/q = 1$. Let $\phi(q) = \left( \int_0^1 |A(xL)/\sqrt{Q/L}|^q dx \right)^{\frac{1}{q}} - \left( \sum_{n=-\infty}^{+\infty} |\eta_n/\sqrt{Q}|^{q'} \right)^{\frac{1}{q'}}$. Since

Fourier series preserve the 2-norm, $\phi(2) = 0$, and since $\phi(q) \geq 0$ for $q \in (1,2)$, we have $\phi'(2) \leq 0$. This translates into

$$\left( \int_0^1 |A(xL)/\sqrt{Q/L}|^2 dx \right)^{\frac{1}{2}} \left\{ -\frac{1}{4} \ln \int_0^1 |A(xL)/\sqrt{Q/L}|^2 dx \right.$$
$$\left. +\frac{1}{2} \frac{\int_0^1 |A(xL)/\sqrt{Q/L}|^2 \ln |A(xL)/\sqrt{Q/L}| dx}{\int_0^1 |A(xL)/\sqrt{Q/L}|^2 dx} \right\}$$
$$- \left( \sum_{n=-\infty}^{+\infty} |\eta_n/\sqrt{Q}|^2 \right)^{\frac{1}{2}} \left\{ \frac{1}{4} \ln \sum_{-\infty}^{+\infty} |\eta_n/\sqrt{Q}|^2 - \frac{1}{2} \frac{\sum_{n=-\infty}^{+\infty} |\eta_n/\sqrt{Q}|^2 \ln |\eta_n/\sqrt{Q}|}{\sum_{n=-\infty}^{+\infty} |\eta_n/\sqrt{Q}|^2} \right\} \leq 0, \tag{41}$$

which yields

$$S(\{\eta_n^2\}) = -\sum_{n=-\infty}^{+\infty} \frac{\eta_n^2}{Q} \ln \frac{\eta_n^2}{Q} \geq \int_0^L \frac{A^2(l)}{Q} \ln \frac{A^2(l)}{Q} dl + \ln L \tag{42}$$

Note that the quantity $\int_0^L \frac{A^2(l)}{Q} \ln \frac{A^2(l)}{Q} dl + \ln L$ is actually independent of $L$ upon re-scaling.

(iii) Next, we show that $S(\{\eta_n\}) \geq S(\{\eta_n^2\})$. Since $\sum_n \eta_n = 1$ and $\eta_n$ is nonnegative, we can always re-order them into a descending series $\hat{\eta}_n$ ($n \geq 0$) with the same entropy as $S(\{\eta_n\})$ (because entropy is permutation-invariant). The result follows as long as $S(\{\hat{\eta}_n\}) \geq S(\{\hat{\eta}_n^2\})$. Let $\beta_n = \hat{\eta}_{n+1}/\hat{\eta}_n \leq 1$. The two series, after normalization, can be written as

$$\begin{aligned} \{\hat{\eta}_n\} &= \hat{\eta}_0,\ \hat{\eta}_0\beta_0,\ \hat{\eta}_0\beta_0\beta_1,\ \hat{\eta}_0\beta_0\beta_1\beta_2,\ \cdots \\ \{\hat{\eta}_n^2/Q\} &= \frac{\hat{\eta}_0^2}{Q},\ \frac{\hat{\eta}_0^2}{Q}\beta_0^2,\ \frac{\hat{\eta}_0^2}{Q}\beta_0^2\beta_1^2,\ \frac{\hat{\eta}_0^2}{Q}\beta_0^2\beta_1^2\beta_2^2,\ \cdots \end{aligned} \tag{43}$$

Consequently,

$$\hat{\eta}_0(1 + \beta_0 + \beta_0\beta_1 + \cdots) = \frac{\hat{\eta}_0^2}{Q}(1 + \beta_0^2 + \beta_0^2\beta_1^2 + \cdots) = 1 \tag{44}$$

This implies that $\hat{\eta}_0 \leq \hat{\eta}_0^2/Q$. Suppose there exists $j \geq 0$ such that

$$\hat{\eta}_0(1 + \beta_0 + \beta_0\beta_1 + \cdots + \beta_0\beta_1\cdots\beta_j) > \frac{\hat{\eta}_0^2}{Q}(1 + \beta_0^2 + \beta_0^2\beta_1^2 + \cdots + \beta_0^2\beta_1^2\cdots\beta_j^2) \tag{45}$$

Then there exists $j' \leq j$ such that $\hat{\eta}_0\beta_0\beta_1\cdots\beta_{j'} > \frac{\hat{\eta}_0^2}{Q}\beta_0^2\beta_1^2\cdots\beta_{j'}^2$. So for any $j \geq j'$, we have $\hat{\eta}_0\beta_0\beta_1\cdots\beta_j > \frac{\hat{\eta}_0^2}{Q}\beta_0^2\beta_1^2\cdots\beta_j^2$. The difference

$$\Delta_j = \hat{\eta}_0(1 + \beta_0 + \beta_0\beta_1 + \cdots + \beta_0\beta_1\cdots\beta_j) - \frac{\hat{\eta}_0^2}{Q}(1 + \beta_0^2 + \beta_0^2\beta_1^2 + \cdots + \beta_0^2\beta_1^2\cdots\beta_j^2) \tag{46}$$

will always be positive and monotonically increases for any $j \geq j'$. This is not consistent with the fact that $\lim_{j\to\infty} \Delta_j = 0$. Thus, we conclude that for any $j$, the partial sum follows the inequality

$$\hat{\eta}_0(1 + \beta_0 + \beta_0\beta_1 + \cdots + \beta_0\beta_1\cdots\beta_j) \leq \frac{\hat{\eta}_0^2}{Q}(1 + \beta_0^2 + \beta_0^2\beta_1^2 + \cdots + \beta_0^2\beta_1^2\cdots\beta_j^2) \tag{47}$$

This is to say that infinite sequence $\{\hat{\eta}_0^2/Q\}$ majorizes $\{\hat{\eta}_n\}$. By Theorem 2.2 of *Li and Busch, 2013*, $S(\{\hat{\eta}_n\}) \geq S(\{\hat{\eta}_n^2\})$, and so $S(\{\eta_n\}) \geq S(\{\eta_n^2\})$

(iv) Finally, since $\zeta_n = \eta_n + \eta_{-n} = 2\eta_n$ for $n \geq 1$, we have

$$S(\{\zeta_n\}) = -\zeta_0 \ln \zeta_0 - \sum_{n \geq 1} 2\eta_n \ln(2\eta_n)$$
$$= -\zeta_0 \ln \zeta_0 - \sum_{n \geq 1} (\eta_n + \eta_{-n})(\ln \eta_n + \ln 2) = S(\{\eta_n\}) - (1 - \eta_0) \ln 2, \tag{48}$$

which is equivalent to

$$S(\{\zeta_n\}) + \left(1 - \frac{1}{L}\int_0^L A(l)\,\mathrm{d}l\right)\ln 2 = S(\{\eta_n\}) \tag{49}$$

Combining previous four steps yields the result of the theorem.

## Entropy associated with the correlation between individuals

### Definition 10 (Entropy of a linear operator)

Let $\mathcal{L}$ be a compact self-adjoint linear operator on a Hilbert space $\mathcal{H}$. If $\mathcal{L}$ has a countable set of eigenvalues $\{\nu_i\}$, and is positive semidefinite, then define the entropy of $\mathcal{L}$ as

$$S(\mathcal{L}) := -\sum_i \frac{\nu_i}{\mathrm{Tr}\mathcal{L}} \ln \frac{\nu_i}{\mathrm{Tr}\mathcal{L}}, \tag{50}$$

where $\mathrm{Tr}\mathcal{L} = \sum_i \nu_i$ is the trace of the linear operator $\mathcal{L}$

### Definition 11 (Mercer's spectrum)

Let $A_j(l_1, l_2)$ be the autocorrelation function for individual $j$ $(1 \leq j \leq J)$, and define the average autocorrelation function as

$$\langle A \rangle(l_1, l_2) = \frac{1}{J}\sum_j A_j(l_1, l_2).$$

Since $A_j$ is Hermitian, the average $\langle A \rangle$ is also Hermitian, thus the integral operator $I_{\langle A \rangle}$ defined by $I_{\langle A \rangle}\phi(l) = \int_0^L \langle A \rangle(l, s)\phi(s)\,\mathrm{d}s$ has a series of real eigenvalues $\{\nu_j\}$ satisfying $\sum_j \nu_j = L$. $\nu_j$ is the solution to the eigenvalue problem:

$$\int_0^L \langle A \rangle(l, s)\phi_j(s)\,\mathrm{d}s = \nu_j \phi_j(l) \tag{51}$$

The spectrum defined by $\{\nu_j/L\}$ is the Mercer's spectrum.

### Definition 12 (Cross-correlation spectrum)

Let the cross-correlation matrix be $\mathbf{C} = \{c_{j,j'}\}_{J \times J}$, where $c_{j,j'}$ is the average cross-correlation between individual $j$ and $j'$:

$$c_{j,j'} = \begin{cases} \frac{1}{L}\int_0^L \mathbf{y}_j^\top(l)\mathbf{y}_{j'}(l)\,\mathrm{d}l & \text{(for the spherical representation)} \\ \frac{1}{L}\int_0^L z_j(l)\overline{z_{j'}(l)}\,\mathrm{d}l & \text{(for the complex representation)} \end{cases} \tag{52}$$

As $\mathbf{C}$ is Hermitian, it has a series of real eigenvalues $\lambda_j$ satisfying $\sum_j \lambda_j = J$. Let the normalized spectrum of $\mathbf{C}$ be $\{\lambda_j/J\}$, then $\{\lambda_j/J\}$ is defined as the cross-correlation spectrum.

The following theorem states that when the complex representation is adopted for bi-ancestry signals, the Mercer's spectrum coincides with the cross-correlation spectrum, so that even if the Mercer's spectrum is calculated using correlation *within* individuals, both captures the correlation *between* individuals.

### Theorem 4

If $\{z_j(l)\}_{j=1}^J$ is a collection of complex bi-ancestry signals, then its Mercers' spectrum is the same as its cross-correlation spectrum.

Proof. The average autocorrelation $\langle A \rangle$ is computed as

$$\langle A \rangle(l_1, l_2) = \frac{1}{J}\sum_j z_j(l_1)\overline{z_j(l_2)}$$

So the integral *equation (51)* becomes

$$\frac{1}{J}\sum_j z_j(l)\int_0^L \overline{z_j(s)}\phi(s)\,\mathrm{d}s = \nu\phi(l),$$

where $(\nu, \phi)$ forms the solution to the above equation. Rearranging the terms, we write

$$\phi(l) = \sum_j \left[\frac{\nu^{-1}}{J}\int_0^L \overline{z_j(s)}\phi(s)\,\mathrm{d}s\right] z_j(l) = \sum_j \alpha_j z_j(l),$$

where $\alpha_j$ stands for the constant inside the bracket. Substituting into the original integral equation, we have

$$\frac{1}{J}\sum_j z_j(l)\int_0^L \overline{z_j(s)}\sum_{j'}\alpha_{j'}z_{j'}(s)\,\mathrm{d}s = \nu\sum_j \alpha_j z_j(l),$$

which is equivalent to

$$\sum_j \left[\sum_{j'}\alpha_{j'}\int_0^L z_{j'}(s)\overline{z_j(s)}\,\mathrm{d}s\right] z_j(l) = J\nu\sum_j \alpha_j z_j(l)$$

Notice that the integral in the bracket contains the cross-correlation between $j'$ and $j$, so if the following relationship holds

$$\sum_{j'}\alpha_{j'}\frac{1}{L}\int_0^L z_{j'}(s)\overline{z_j(s)}\,\mathrm{d}s = \sum_{j'}\alpha_{j'}\overline{c_{j,j'}} = \frac{J\nu}{L}\alpha_j$$

then the system has a solution of $\nu$. The above equation is equivalent to

$$\overline{\mathbf{C}}\alpha = \frac{J\nu}{L}\alpha \Leftrightarrow \mathbf{C}\overline{\alpha} = \frac{J\nu}{L}\overline{\alpha}$$

Which means that $J\nu/L$ is the eigenvalue of the cross-correlation matrix $\mathbf{C}$, and since $\mathbf{C}$ is a Hermitian matrix, the existence of its real eigenvalues is guaranteed. So for each $\nu_i$, we have $\lambda_j = J\nu_j/L$, which is to say

$$\frac{\lambda_j}{J} = \frac{\nu_j}{L}, \quad \text{for } \forall i$$

As both spectra coincide for the complex representation, we can define the between-individual entropy using either of them.

## Definition 13 (Between-individual entropy of the complex representation)

For complex bi-ancestry signals, let $S_b := S(I_{\langle A\rangle}) = S(\mathbf{C})$. $S_b$ is the Shannon entropy characterizing the between-individual correlation.

Corollary 1 (Maximum $S_b$).

For $J$ diploid individuals with the complex representation of ancestry, the largest attainable $S_b$ is given by

$$\begin{aligned}
S_{b,\max} &= -\frac{(J-1)(1-c)}{J}\ln\frac{1-c}{J} - \frac{1+(J-1)c}{J}\ln\frac{1+(J-1)c}{J}, \\
&\approx (1-c)\ln J - (1-c)\ln(1-c) - c\ln c \quad (J\gg 1)
\end{aligned}$$

(53)

where $c = (3 + 2\sqrt{2})/8$.

Proof. For entropy to be large, autocorrelation within each individual must be very weak. This will occur if a sufficient amount of recombination has taken place such that the ancestry across the genome is largely independent between loci. The maximum value will occur when both parental lineages contribute equally to the hybrid ancestry, so that the hybrid index $h = 0.5$. The heterozygosity $H$ is then 0.5 following a random distribution of ancestries along the two genome copies. The off-diagonal elements in the cross-correlation matrix $\mathbf{C}$ thus take the value

$$c = 2 \times \left(\frac{1}{2} + \frac{\sqrt{2}-1}{2}\frac{1}{2}\right)^2 = \frac{3+2\sqrt{2}}{8}$$

For a $J \times J$ matrix whose diagonal elements are all 1, and whose off-diagonal elements are all $c$, its normalized eigenvalues are given by $\{\frac{1+(J-1)c}{J}, \frac{1-c}{J}, \cdots, \frac{1-c}{J}\}$, which leads to the above result.

Corollary 2 (Minimum $S_b$). *$S_b = 0$ is the minimum between-individual entropy, and is attainable if recombination is completely suppressed in the genomic interval of interest.*

Proof. If recombination is completely suppressed along $[0, L]$, the ancestry signal is constant along the interval in any individual with any ploidy. Hence, $A_j \equiv 1$, and $\langle A \rangle \equiv 1$. The only non-zero eigenvalue associated with a constant integral kernel is $\nu = L$, so that we have $S_b = 0$ using the Mercer's spectrum. Note that the converse is not true, because the diploid ancestry signal is unaware of phase. A region completely heterozygous may in fact has a recombination break point, and the ancestry can flip phases when crossing the break point. However, this extreme situation is unlikely as it requires two separate recombination events to have occurred at precisely the same point and in opposite directions. If the hybrid zone is old, a long track of heterozygous ancestry is therefore usually good evidence for the presence of barrier loci.

