## [Editor Report]

The authors leverage theory, simulations, and empirical population genomics to evaluate what are the consequences of differences in substitution rates in hybridizing species. This is a largely overlooked pheonomenon. This study highlights the issue and demonstrates that two hybridizing species of Papilio have differences in thir substitution rates. The work will be of interest to a large group of evolutionary biologists, especially those studying evolution at the whole-genome level.

---

## [Decision Letter]

**Decision letter after peer review:**

Thank you for submitting your article "Admixture of evolutionary rates across a hybrid zone" for consideration by *eLife*. Your article has been reviewed by 2 peer reviewers, and the evaluation has been overseen by a Reviewing Editor and George Perry as the Senior Editor. The following individual involved in review of your submission has agreed to reveal their identity: Daniel L Powell (Reviewer #1).

Essential revisions:

Overall, the manuscript does a good job of presenting results in a clear and concise manner. Before making a final decision I have three requests:

1. The reviewers have requested some clarifications to make the entropy derivations more accessible to the reader, soften some claims (i.e., archaic gene flow), and add a section regarding the caveats to the discussion. I strongly agree with the reviewers and a potential resubmission should include a response to these requests.

I have two more things to add.

2. Please add the study organisms to the title of the manuscript. As it is currently written, the title is overly ambitious and the reader might be misled to think that the differences in substitution rates is a general phenomenon.

3. The scholarship of the piece needs to be improved. In the current form, the discussion has seven references which is rather surprising given that the study of introgression is one of the most vibrant aspects of current speciation research. The discussion needs to be more encompassing of our current understanding of the factors that determine the fate of an introgressed allele. The current form of the discussion should elaborate on what are the primary hypothesis that could lead to differences in substitution rates and assess what are the potential implications for the interaction between selection and recombination (in the face of differences in substitution rates).

*Reviewer #1 (Recommendations for the authors):*

I have nothing in the way ion further suggestions for the authors that isn't included in the public review except the following.

Lines 187-188: In the previous paragraph you claim that divergent rates likely exist, but here you state that you've established that they do. This strengthening of the statement seems unwarranted.

*Reviewer #2 (Recommendations for the authors):*

In their manuscript ("Admixture of evolutionary rates across a hybrid zone"), Xiong et al. use whole genome resequencing data to assess rates of genome evolution between two species of butterflies and determine whether loci that evade introgression between the species are also those that evolve at asymmetric rates. They show the landscape of divergence is heterogeneous among these sister species and the genomes of the hybrids exhibit correlated ancestry blocks (both within an individual's genome, and between individuals). Interestingly, sites that are more strongly diverged appear to be more tightly correlated, indicative of potential barrier loci. The authors also find significantly skewed D statistics using four outgroups. While these statistics might normally be used to infer introgression, the authors further use topological weights, and node heights to show that asymmetric rates of genome evolution might account for these skewed D statistics, rather than introgression. Using both coalescent theory and empirical observations, they show that these asymmetrically evolving sites are correlated with variation in relative divergence among species. Overall, I congratulate the authors on a clever and well written manuscript. I think this work is incredibly rigorous and I appreciate the clear communication of fairly complex results. I do have a few suggestions, which I think the authors might find useful.

Namely, while the authors nicely show potential asymmetric evolutionary rates between species, and also show that highly divergent sites are more likely to show patterns of lower entropy (i.e. less randomly associated with other loci), which I agree may be indicative of barrier loci, I feel there might be a missing link between these two processes in the way the manuscript is currently written. For example, the authors show these differences in evolutionary rates (i.e. the ratio number of substitutions between lineages) is correlated with divergence, but I'm curious if they are also correlated with ancestry entropy. In my mind, I agree with the authors that barrier loci should be highly diverged (and exhibit reduced diversity), but these signatures can be caused by a number of evolutionary processes. Instead, the entropy data is quite striking to me and more convincing of barrier loci. Therefore, I might expect the entropy measurements to also be correlated with the D statistics or the ratio of substitution rates.

I find Figures 4 and 6 really convincing of differential evolutionary rates over differences in introgression with the four outgroups. One additionally might predict that if two species were evolving differentially (and this in turn was reflected in the relative divergence between these sister species), that one might also see asymmetric differences in something like DXY between each focal species and each outgroup, and this might also be correlated with FST between the two sister lineages. This might be nice as a supplement to bolster your argument.

Lastly, I think the authors could clarify two important aspects of their differential evolutionary rate hypothesis.

First: it's unclear to me whether the authors are arguing that barrier loci evolve at different rates than the rest of the genome (and therefore show both increased divergence and increased asymmetric rates of evolution), or whether loci that evolve at different rates may have a harder time introgressing between species (by nature of the fact that a given locus' evolutionary rate is at odds with the evolutionary rate of the rest of the recipient genome). Perhaps this is a difference without much distinction, but a bit more detail from the authors on this might be useful. Additionally, if the authors are arguing the former, I think there's a nice amount of literature that they could tap into to talk about why different barrier loci might evolve rapidly (for example, if they are involved in a genetic arms race).

Second: The finding that these two sister species have such drastically different evolutionary rates is really interesting, and I think the authors could elaborate a bit more on why they think that is. There is some explanation on lines 232-234, but I'm curious whether one of these species has undergone a massive bottle neck or partakes in more inbreeding in ways that are also known to affect D statistics and such.

---

## [Author Response]

1. The reviewers have requested some clarifications to make the entropy derivations more accessible to the reader, soften some claims (i.e., archaic gene flow), and add a section regarding the caveats to the discussion. I strongly agree with the reviewers and a potential resubmission should include a response to these requests.

We thank both reviewers and the reviewing editor for their encouraging assessments of this manuscript. All their suggestions have helped us state the results in a more accurate and meaningful way.

2. Please add the study organisms to the title of the manuscript. As it is currently written, the title is overly ambitious and the reader might be misled to think that the differences in substitution rates is a general phenomenon.

“Butterfly” is now added to the title. (“Admixture of evolutionary rates across a butterfly hybrid zone”)

3. The scholarship of the piece needs to be improved. In the current form, the discussion has seven references which is rather surprising given that the study of introgression is one of the most vibrant aspects of current speciation research. The discussion needs to be more encompassing of our current understanding of the factors that determine the fate of an introgressed allele. The current form of the discussion should elaborate on what are the primary hypothesis that could lead to differences in substitution rates and assess what are the potential implications for the interaction between selection and recombination (in the face of differences in substitution rates).

To improve the scholarship and discuss these results in light of existing and historical work, we have added new references and carried out a major revision of the Discussion, with the following subsections:

1) Entropy as a useful measure of ancestry randomness. We discuss the incentives behind developing these metrics, and the intuition for choosing particular forms of ancestry representation. We also explain the impact of other processes such as inbreeding and recombination on entropy, and show how they are useful for window-based genomic analysis.

2) Potential mechanisms of divergent substitution rates. Here, we speculate on several ways that different substitution rates might result. Ohta’s nearly neutral theory is now included as a possible explanation for smaller asymmetric rates on nonsynonymous substitutions than synonymous and intronic ones.

3) Are introgression tests robust to substitution rate variation? This part summarizes the finding that introgression tests based on site patterns can produce false-positives when substitution rates diverge between species and recurrent mutations exist in outgroups. In particular, we stress that, since barrier loci have the most asymmetric rates of evolution during hybridization (given the existence of divergent rates), they might be falsely identified as having been introgressed from an outgroup when introgression tests are used on genomic windows. This result might therefore mislead researchers to the conclusion that some “speciation genes” are borrowed from outgroups.

4) The conceptual picture of rate-mixing. In this final subsection, we recapitulate the main conclusion of this study, that locally asymmetric substitution rates reflect loci responsible for reproductive isolation. The caveats of the theory we develop in this manuscript are discussed in detail, and we conjecture that the monotonic relationship between rate asymmetry and sequence divergence might be a robust signature irrespective of model details.

Reviewer #1 (Recommendations for the authors):I have nothing in the way ion further suggestions for the authors that isn't included in the public review except the following.Lines 187-188: In the previous paragraph you claim that divergent rates likely exist, but here you state that you've established that they do. This strengthening of the statement seems unwarranted.

Thanks! We changed the first sentence of this subsection to “Having tested for …” rather than “Having established the existence of …”

Reviewer #2 (Recommendations for the authors):In their manuscript ("Admixture of evolutionary rates across a hybrid zone"), Xiong et al. use whole genome resequencing data to assess rates of genome evolution between two species of butterflies and determine whether loci that evade introgression between the species are also those that evolve at asymmetric rates. They show the landscape of divergence is heterogeneous among these sister species and the genomes of the hybrids exhibit correlated ancestry blocks (both within an individual's genome, and between individuals). Interestingly, sites that are more strongly diverged appear to be more tightly correlated, indicative of potential barrier loci. The authors also find significantly skewed D statistics using four outgroups. While these statistics might normally be used to infer introgression, the authors further use topological weights, and node heights to show that asymmetric rates of genome evolution might account for these skewed D statistics, rather than introgression. Using both coalescent theory and empirical observations, they show that these asymmetrically evolving sites are correlated with variation in relative divergence among species. Overall, I congratulate the authors on a clever and well written manuscript. I think this work is incredibly rigorous and I appreciate the clear communication of fairly complex results. I do have a few suggestions, which I think the authors might find useful.Namely, while the authors nicely show potential asymmetric evolutionary rates between species, and also show that highly divergent sites are more likely to show patterns of lower entropy (i.e. less randomly associated with other loci), which I agree may be indicative of barrier loci, I feel there might be a missing link between these two processes in the way the manuscript is currently written. For example, the authors show these differences in evolutionary rates (i.e. the ratio number of substitutions between lineages) is correlated with divergence, but I'm curious if they are also correlated with ancestry entropy. In my mind, I agree with the authors that barrier loci should be highly diverged (and exhibit reduced diversity), but these signatures can be caused by a number of evolutionary processes. Instead, the entropy data is quite striking to me and more convincing of barrier loci. Therefore, I might expect the entropy measurements to also be correlated with the D statistics or the ratio of substitution rates.

We agree with the intuition of the reviewer here that entropy could also be correlated with rate asymmetry. In fact, since the measure of "divergence" we use, F_ST_. is negatively correlated with entropy (Figure 3E, third panel), and also positively correlated with rate ratio r, it is likely that entropy will be negatively correlated with r. The reason we use F_ST_ instead of entropy in studying the process of rate-mixing between populations is as follows:

First, F_ST_ is widely used in comparing populations, and has well-understood properties in population genetics theory. Yes, when F_ST_ is used to study gene flow, results can be misled by a number of other evolutionary processes (e.g., linked selection not caused by barrier loci), entropy is not exempt from complications either. As stated in the response to Reviewer #1, inbreeding reduces between-individual entropy (S_b_), and low recombination rates reduce within-individual entropy (S_w_), and both processes are not directly related to barrier loci per se.

Second, as entropy is defined on estimated local ancestry rather than DNA sequences, its performance will depend on the statistical power of ancestry estimation, which might introduce an additional layer of uncertainty to the results. However, F_ST_ can be directly estimated from sequences.

Third, we here use the entropy measure only within the hybrid population, while F_ST_ could be calculated between any populations (in this case, the “pure” populations), and so is potentially applicable to a wider set of data in which hybrid populations have not been sampled.

I find Figures 4 and 6 really convincing of differential evolutionary rates over differences in introgression with the four outgroups. One additionally might predict that if two species were evolving differentially (and this in turn was reflected in the relative divergence between these sister species), that one might also see asymmetric differences in something like DXY between each focal species and each outgroup, and this might also be correlated with FST between the two sister lineages. This might be nice as a supplement to bolster your argument.

Yes! Interestingly, D_XY_-based statistics were the very first clues we had that showed some kind of asymmetry between these two species, and we were then genuinely puzzled by this asymmetry, which initially was thought to show introgression with an outgroup (we only included one outgroup *P. bianor*, at the early stage of the analysis).

The statistic we used originally was “Population Branch Statistic” (PBS). In one version of the PBS, choose a triplet (P1, P2, O). P1 and P2 are two focal taxa, while O is an outgroup. PBS describes how distant each lineage is from the center of the triplet, and is calculated as follows:PBS(P1)=log⁡11−DXY(P1,O)+log⁡11−DXY(P1,P2)−log⁡11−DXY(P2,O)2PBS(P2)=log⁡11−DXY(P2,O)+log⁡11−DXY(P1,P2)−log⁡11−DXY(P1,O)2

(see: https://github.com/thamala/PBScan/blob/master/PBScan_manual.pdf)

Many people use PBS to find regions with unusually fast evolution in one lineage (indicating selection). In our case, among 50kb sliding windows, PBS(*maackii*) is usually larger than PBS(*syfanius*), and this asymmetry increases with F_ST_. For instance, chromosome 15 has one of the lowest F_ST_, while chromosome 30 (Z) has the highest F_ST_, and PBS is highly skewed towards *P. maackii* on chromosome 30.

**Author response image 1. sa2fig1:** 

We then thought that such a large skew in PBS couldn’t all be attributed to natural selection in *P. maackii*, and that there must be other processes generating such patterns. However, because PBS is not a normalized measure, it’s difficult to make a fair comparison across genomic windows. For this reason, we switched to D_3_ and D_4_ statistics, both are normalized, and are specifically designed to capture asymmetry information.In short, it is absolutely true that information based on D_XY_ will show similar asymmetry between the focal taxa. But we argue that the normalized measures using D statistics make the point more clearly.

Lastly, I think the authors could clarify two important aspects of their differential evolutionary rate hypothesis.First: it's unclear to me whether the authors are arguing that barrier loci evolve at different rates than the rest of the genome (and therefore show both increased divergence and increased asymmetric rates of evolution), or whether loci that evolve at different rates may have a harder time introgressing between species (by nature of the fact that a given locus' evolutionary rate is at odds with the evolutionary rate of the rest of the recipient genome). Perhaps this is a difference without much distinction, but a bit more detail from the authors on this might be useful. Additionally, if the authors are arguing the former, I think there's a nice amount of literature that they could tap into to talk about why different barrier loci might evolve rapidly (for example, if they are involved in a genetic arms race).

Thank you for point it out. We apologize for not being clear on this result. The main idea throughout the manuscript is simply that:

1. DNA accumulates substitutions at different rates between the two species

2. Hybridization mixes these rates for any specific locus, but at different rates of mixture depending on proximity to barrier loci.

To improve the wording, we now always use “mix” instead of “couple” whenever we are talking about rates being mixed between the two species. In the abstract, we also emphasize this mixing process is “between species”, and so we are not comparing “between loci”.

Second: The finding that these two sister species have such drastically different evolutionary rates is really interesting, and I think the authors could elaborate a bit more on why they think that is. There is some explanation on lines 232-234, but I'm curious whether one of these species has undergone a massive bottle neck or partakes in more inbreeding in ways that are also known to affect D statistics and such.

There is now a subsection in Discussion to include possible mechanisms of substitution rate differences. Although it is somewhat speculative, we believe it may be to do with the different numbers of generations per year between the two species. The highland lineage has 1-2 generations a year while the lowland lineage has 2-3. If everything else is equal, the ratio between the generation numbers could well be about ~1.8. We also speculate that Ohta’s nearly neutral theory might explain the reduced rate asymmetry of ~1.6 for nonsynonymous sites (the lowland lineage has larger populations more efficient selection against mildly deleterious mutations slowing down nonsynonymous substitutions more in the lowland).